# Finite Mixture Models in the Evaluation of Positional Accuracy of Geospatial Data

José Rodríguez-Avi [1,*,†]  and Francisco Javier Ariza-López [2,†] 

1 Departamento de Estadística e Investigación Operativa, Universidad de Jaén, 23071 Jaén, Spain
2 Departamento de Ingeniería Cartográfica, Geodésica y Fotogrametría, Universidad de Jaén, 23071 Jaén, Spain; fjariza@ujaen.es
* Correspondence: jravi@ujaen.es; Tel.: +34-638-925-891
† These authors contributed equally to this work.

**Abstract:** Digital elevation models (DEMs) are highly relevant geospatial products, and their positional accuracy has demonstrated influence on elevation derivatives (e.g., slope, aspect, curvature, etc.) and GIS results (e.g., drainage network and watershed delineation, etc.). The accuracy assessment of the DEMs is usually based on analyzing the altimetric component by means of positional accuracy assessment methods that are based on the use of a normal distribution for error modeling but, unfortunately, the observed distribution of the altimetric errors is not always normal. This paper proposes the application of a finite mixture model (FMM) to model altimetric errors. The way to adjust the FMM is provided. Moreover, the behavior under sampling is analyzed when applying different positional accuracy assessment standards such as National Map Accuracy Standards (NMAS), Engineering Map Accuracy Standard (EMAS) and National Standard for Spatial Data Accuracy (NSSDA) under the consideration of the FMM and the traditional approach-based one-single normal distribution model (1NDM). For the NMAS, the FMM performs statistically much better than the 1NDM when considering all the tolerance values and sample sizes. For the EMAS, the type I error level is around 3.5 times higher in the case of the 1NDM than in the case of the FMM. In the case of the NSSDA, as it has been applied in this research (simple comparison of values, not hypothesis testing), there is no great difference in behavior. The conclusions are clear; the FMM offers results that are always more consistent with the real distribution of errors, and with the supposed statistical behavior of the positional accuracy assessment standard when based on hypothesis testing.

**Keywords:** DEM; normal distribution; parametric error models; Gaussian finite mixture models

## 1. Introduction

Positional accuracy has always been considered a defining and essential element of the quality of any geospatial data [1], as it affects factors such as geometry, topology, and thematic quality; it is directly related to the interoperability of spatial data [2]. Considering the widespread use of geospatial information and the interoperability requirements of different geomatics applications and spatial data infrastructures (SDIs), it is crucial to ensure information quality, as this is the only means of guaranteeing reliable solutions when making decisions [3]. A particular case of geospatial data is that of digital elevation models (DEMs). Currently, there are numerous technologies (GNSS, LiDAR, InSAR, etc.) [3,4], which allow the generation of DEM data products with very diverse characteristics (numerical precision, spacing, grid storage, etc.) [3,5]. DEMs are a key data type for many applications domains because they provide the height component in GIS analysis, the geomorphological description of the land [6], which is a reference surface for all hydrological applications (water cycle, erosion, floods, etc). In [7], the basis for the development of forestry models [8] and the base for agricultural parcel rating [9] is useful in every analysis task related to civil engineering [10]. DEMs are part of the information infrastructure to achieve the Sustainable

Development Goals and are considered as Global Fundamental Geospatial Themes by the United Nations [11]; they are also included in the list of geospatial themes of the European Spatial Infrastructure [12]. The data model most used in the case of DEMs is the grid [13,14]. Usually, in the case of gridded DEMs, the evaluation of positional accuracy is limited to the errors in the altimetric component (elevation/height) (Case 1D). This 1D perspective is of interest in this document, since, without loss of generality, it allows a simpler approach to the proposed method. The positional accuracy in DEMs has a direct influence on elevation derivatives such as slope, aspect and curvature, and generates erroneous drainage network or watershed delineation [15,16]. Vertical positional accuracy requirements depend on the scale and specific use case; in this line, [15,17] present indicative accuracy values for some usual DEM applications.

Positional accuracy assessment methods (PAAMs) are standardized processes to either estimate or control the positional quality [18] of geospatial data. The PAAMs understand the quality of the data product as the presence of errors with a limited size (e.g., lesser than a tolerance value for the bias or for the dispersion). The accuracy estimation consists of determining a reliable value of the property of interest (e.g., mean bias, standard deviation, proportion, etc.), in the data product. These methods provide a value and its corresponding confidence interval as a result (e.g., a mean value and its deviation such as 5.27 m ± 0.15 m). On the other hand, quality control involves deciding whether or not the property of interest in that data product reaches a certain quality level. These are intended to provide a statistical basis for making an acceptance/rejection decision as a consequence of compliance/noncompliance with a specification (e.g., given the specification that no more than 5% of the elements present 1D-positional errors greater than 1 m, a decision is made to accept/reject according to the evidence found in the sample). In this sense, specific recommendations for the positional assessment of DEM can be observed in [18].

Acquisition technologies used in the positional accuracy assessment, such as Global Navigation Satellite Systems (GNSS) and LiDAR systems, enable the collection of coordinates in the field with high accuracy, which increases the possibility of more accurate positional accuracy assessments. Moreover, PAAMs have evolved over time, from the National Map Accuracy Standard (NMAS) [19] to the more recent by the American Society for Photogrammetry and Remote Sensing, called the Positional Accuracy Standards for Digital Geospatial Data [20], in which the statistics are based on the National Standard for Spatial Accuracy (NSSDA) [21]. It should be noted that these PAAMs apply to both planimetric control (2D-error data) and altimetric control (1D-error data). It is interesting to analyze these three PAAMs, as they present different and complementary perspectives. The NMAS can be considered a method with capabilities to work with free-distributed data [21]. This standard sets out a method of positional accuracy control that establishes an acceptance/rejection rule in a very simple manner, and is based on the binomial distribution applied to error counts. This standard is outdated, however, as it refers to tolerances defined on paper, that is, to the representation scale, but its conceptual basis can be applied to any tolerance value. The Engineering Map Accuracy Standards [22] assumes that positional errors are normally distributed and proposes a set of statistical hypothesis tests that must be overcome for the product to be accepted. Specifically, it establishes two statistical tests per component, one focused on the detection of biases (Student's *t*-test) and the other on the behavior of dispersion (Chi square test). Finally, the NSSDA assumes the normality of the error data and is not a positional accuracy control method, as it does not establish acceptance or rejection; the result is a value and, therefore, is an estimation method.

The normal distribution function remains the theoretical base model for some widely used PAAMs (e.g., for the EMAS and the NSSDA) because it is a suitable distribution for representing real-valued random variables generated purely at random. In fact, what is desirable when working with measurement errors is their normal distribution, as this implies that there are no other unknown causes—which are therefore uncontrollable—that affect the measurement result. But, in practice, it is hard to find error data sets that, strictly, could be adequately modeled with one normal distribution. This circumstance has been

highlighted specially for the case of DEM [23,24]. This can be due to various causes that can appear alone or together (e.g., many extreme values, overlap of several processes, elimination of data, distribution of values closes to zero or the natural limit, and so on). For these reasons, alternatives based on robust statistics [25]), nonparametric models such as the observed distribution [26], on error counting [27] or percentiles [28], among others, have been proposed. Therefore, we have chosen the case applied to DEMs because it offers a situation where the non-normality of the errors has already been indicated in previous studies and because dealing with 1D errors is a simpler situation than the case of 2D errors, which makes it easy to explain.

In this paper, we explore the case when, even assuming underlying normality, errors come from different normal distributions, that is to say, normal distributions with different parameters. In this case, an approach based on the use of Gaussian finite mixture models (FMM) is adequate for obtaining a whole parametric model that reproduces the empirical distribution of observed data [29–32]. This approach to the problem is chosen because the FMMs are nothing more than the extension of the traditional model based on a one-single normal distribution. This offers the user a familiar framework with the advantages of a parametric model for statistical inference questions. In addition, FMMs offer enough robustness and adaptability to particular distributions that can demonstrate the very varied possible use cases.

In this work, a double objective is pursued. Firstly, to study the distribution of the estimators in the sampling under the FMM, which allows proposing specific hypothesis tests for the fitted model, and secondly, to apply this study to various positional accuracy standards (specifically NMAS, EMAS and NSSDA) , for which the theoretical framework is defined, the procedure is developed and it is verified how its use improves the results obtained under the assumption of a single normal distribution. Therefore, our ultimate goal is just to propose a parametric model that can replace the normal univariate statistical model (widely accepted and applied) and that can be used in all cases that are required, but not to develop a new model (theoretical or empirical) for the uncertainty or new specific indices for the evaluation of positional accuracy.

After this section, the conceptual bases of the finite mixture model are presented. In Section 3, an overview of the methods is presented, which includes the adjustment process of the FMM and the simulation process to analyse the behavior when applied to the selected PAAMs. Section 4 presents the data; these are altimetric discrepancies from two digital terrain models. Section 5 shows the results obtained and the application to the different standards.It is long because it presents the results of the FMM adjustment process and also of the simulation process for the three PAAMs under analysis. The Sections 6 and 7 are devoted to presenting the discussions and conclusions.

## 2. Finite Mixture Models

This article proposes the application of the finite Gaussian mixture model methodology to fit a set of measurement errors. A detailed analysis can be observed in [29–32] and may be summarized as follows:

- Let the vector of observed errors $X = (X_1, \ldots, X_n)$, a random sample that comes from a mixture of $g > 1$ distributions $\Phi_i = \mathcal{N}(\mu_i, \sigma_i), i = 1, \ldots, g$, in the way that each of which appears with a proportion $\pi_i$ in the mixture, $\pi_1 + \cdots + \pi_g = 1$. Then, the value of the density function of each $X_i$ is given by:

$$f_\theta(x_i) = \sum_{j=1}^{g} \pi_j \phi_j(x_i); \qquad x_i \in \mathbb{R} \tag{1}$$

- Which implies estimating the vector of parameters

$$\Theta = ((\pi_1, \mu_1, \sigma_1), \ldots, (\pi_g, \mu_g, \sigma_g)) \tag{2}$$

of dimension $3g$.

- The estimation of $\boldsymbol{\Theta}$ (2) is made with the *EM* algorithm [30,33–35], which is obtained iteratively through the operator

$$Q\left(\theta|\theta^{(t)}\right) = E\left[\log h_\theta(C)|x, \theta^{(t)}\right] \tag{3}$$

where $\theta \in \boldsymbol{\Theta}$, $\theta^{(t)}$ is the value of the iteration $t$ and the expectation refers to the distribution of $k_\theta(c|x)$ of $c$ given $x$ for the value $\theta^{(t)}$ of the parameter.

- In this way, $g$ groups are calculated. The posterior probability of pertaining to the group $i, i = 1, \ldots, g$ is given by

$$\hat{\pi}_{ij} = \frac{\hat{\pi}_i f_i\left(x_j|(\hat{\mu}_i, \hat{\sigma}_i)\right)}{\sum_{k=1}^g \hat{\pi}_k f_k(x_k|(\hat{\mu}_k, \hat{\sigma}_k))}; \qquad x_j \in \mathbb{R}, i = 1, \ldots, g; j = 1, \ldots, n \tag{4}$$

and each sample point $x_j$ is assigned to the group where $\hat{\pi}_{ij}$ is maximum.

- The final density function is:

$$f(x_j) = \sum_{i=1}^g \hat{\pi}_{ij} \tag{5}$$

where $\hat{\pi}_{ij}$ are obtained in (4).

In order to determine the best value of $g$ (the final number of mixing distributions), the use of some information criteria to choose the best fitted model is proposed. In this case, they are the Akaike Information Criteria, *AIC* and the Bayesian Information Criteria, *BIC* (see for instance [36,37]):

$$AIC_g = -2\mathcal{L}_{\}} + 2p \tag{6}$$
$$BIC_g = -2\mathcal{L}_{\}} + p\ln(n) \tag{7}$$

where $\mathcal{L}_{\}}$ is the log-likelihood value in the estimation with $g$ groups and $p = 3g$ is the number of estimated parameters (2). In both cases, the best value of $g$ corresponds to the one in which the value obtained by *AIC* or *BIC* is the minimum. The difference between both measures is the presence in the *BIC* of the sampling size $n$ in order to correct the criterion value. This criterion penalizes models with a greater number of estimated parameters by replacing the term "$2p$" by "$p\ln(n)$", thus obtaining models of lower order than those obtained by the *AIC*, which allows for correcting the tendency to overestimate. To implement the calculations, the package *mixtools* of R [38,39] has been employed.

Once selected, the theoretical model provides a whole description about the population where data come from, and all population probabilities and parameters can be calculated. In this case:

- Mean:

$$\hat{\mu} = \sum_{i=1}^g \hat{\pi}_i \hat{\mu}_i \tag{8}$$

- Variance:

$$\hat{\sigma}^2 = \sum_{i=1}^g \hat{\pi}_i \hat{\sigma}_i^2 + \sum_{i=1}^g (\hat{\mu}_i - \hat{\mu})^2 \tag{9}$$

and, in consequence, $\sigma = \sqrt{\sigma^2}$

## 3. Methods

Two well-differentiated parts can be considered:

1. Estimating of a model based on mixtures (Section 3.1). This step will offer the parameters of the mixing distribution functions (proportions, means and deviations). In this way, a parametric model based on the mixture of normal distributions will be available.
2. Simulation of the behavior of PAAMs in sampling processes (Section 3.2). By means of the simulation of samples it will be known how the estimates of the variables used by PAAMS (e.g., mean and standard deviation) behave when a parametric model based on a finite mixture of normal distributions is applied in comparison with the traditional approach based on one normal distribution model.

The next two subsections describe these two parts in more detail and set out the proposed methodology for their use.

### 3.1. Estimating the Finite Mixture Model

The steps for obtaining this model are:

- To take a sufficiently representative sample.
- To adjust several mixing models with different finite numbers of mixed normal distributions (e.g., 2, 3, 4 and so on).
- To determine the "best fitted" mixing model.

In relation to the first step, the utility of the resulting mixing model is depending on the representativeness of the sample used for its adjustment. This work uses the whole error model (discrepancies) of Section 4 dedicated to describing the data. In this way, the representativeness of the results is assured for this area.

The second step consists on selecting $g$, which is the number of normal mixing distributions that provides the best fit according to $AIC$ or $BIC$ values. Once selected, the third step consist on studying the model density function (5) with the selected value of parameters (2), and to compare it with the observed data.

### 3.2. Simulation of the Behavior of PAAMs in Sampling Processes

PAAMs (e.g., NMAS, EMAS, NSSDA, etc.) applied to geospatial data products are based on samples from which one or several parameters (e.g., a proportion, the mean, the standard deviation) are derived. Some of these parameters are used for defining a classical one-single normal distribution model approach (e.g., $\mathcal{N}(\mu, \sigma)$), which is used by several PAASMs (e.g., EMAS, NSSDA). Therefore, it is interesting to know the distribution in the sampling of these parameters, and compare their behavior under two approaches: (i) the model based on a one-single normal distribution (1NDM), and (ii) the Finite Mixture Model (FMM), fixed following the process indicated in Section 3.1.

Once the parameters of the FMM have been obtained, we are interested in determining its behavior in sampling processes. In this sense, a Montecarlo simulation will be carried out. The process consists of generating random samples of different sizes and determining their quantiles for each one of them under the two approaches. The considered sample sizes are $n \in [20, 30, 40, 50, 80, 100, 200, 500]$, and 5000 iterations are performed in the simulation. For each simulation, and for each sample size, the sample mean and variance are calculated, resulting in a vector of 5000 means and standard deviations, which may be considered as a sample of the sampling distribution of the estimators of the model. Through these simulations, the distribution of the estimators is estimated and the quantiles to be used are determined (for example, 5% or 1%). These quantiles will be later used to obtain critical values for tests. Figure 1 shows a general view of this simulation process.

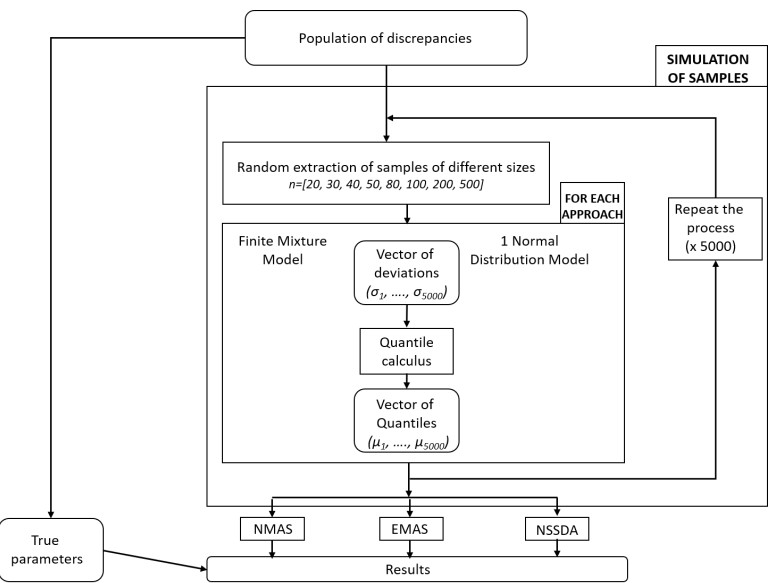

**Figure 1.** The simulation process for the comparison of the analyzed positional accuracy assessment approaches.

## 4. Discrepancy Data for the Application Case

In order to simplify the example case, $1D$-positional-error data are used. In any case, the process shown here is valid for all PAAMs that consider the components of the horizontal positional error ($e_x$ and $e_y$) as one-dimensional normal variables. In this study case, the errors are vertical and the $1NDM$ and $FMD$ models will be applied to discrepancy data (errors) obtained in a study area around Allo (Navarra, Spain). It is a mid-mountain area of 504 km$^2$, where the elevation varies between 316 and 1046 m; the average elevation is 468 m and the standard deviation of elevations is 92.8 m. A map of the studied area appears on Figure 2.

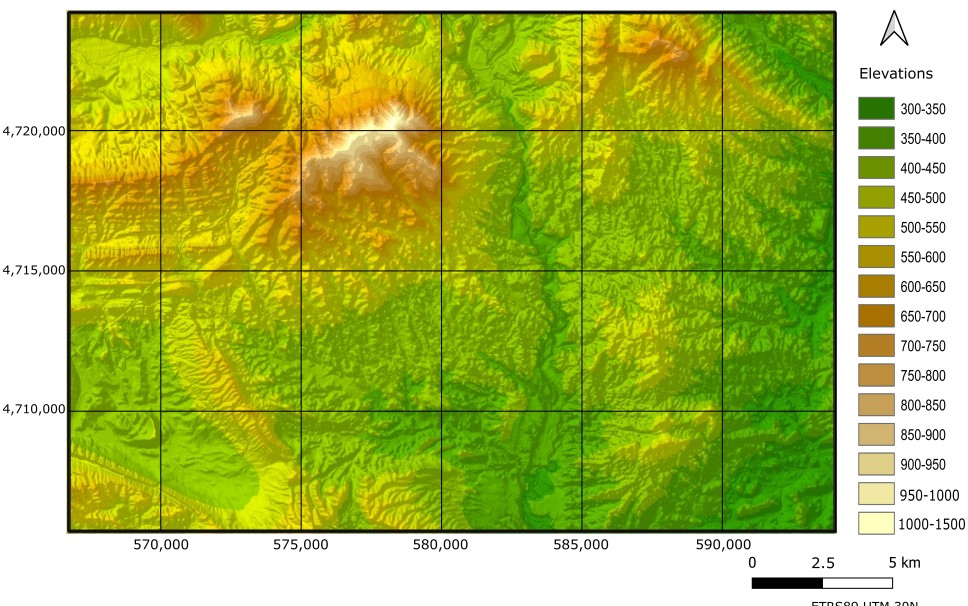

**Figure 2.** Map of the zone of Allo in Navarra (Spain).

Discrepancy is derived as the difference between two DEMs:

$$d_i = h_{DEM,i} - h_{REF,i} \tag{10}$$

where

- $h_{DEM,i}$: elevation in position $i$ of a DEM product;
- $h_{REF,i}$: elevation in position $i$ of a reference;
- $d_i$: discrepancy in elevation in position $i$.

  In this study, the DEM data sets are:

- *REF* (Reference): DEM02. In this case, it is a gridded DEM ($2 \times 2$ m resolution). Its primary data source is an aerial LiDAR survey obtained in 2017 (second coverage of the PNOA-LiDAR project https://pnoa.ign.es/estado-del-proyecto-lidar/segunda-cobertura, accessed on 28 March 2022). The informed positional accuracies for the DEM are $RMSE_{XY} \leq 50$ cm and $RMSE_{Z02} \leq 25$ cm.
- *DEM* (Product): DEM05 is a gridded DEM ($5 \times 5$ m resolution) that comes from an aerial LiDAR survey obtained in 2012 (first coverage of the PNOA-LiDAR project https://pnoa.ign.es/estado-del-proyecto-lidar/primera-cobertura, accessed on 28 March 2022). The informed positional accuracies for the DEM are $RMSE_{XY} \leq 50$ cm and $RMSE_{Z05} \leq 50$ cm.

Both data sets can be considered independent in their generation. However, the one used as a reference (DEM02) does not meet the criteria of being a true reference because its accuracy is not at least three times better than that of the product to be evaluated (DEM05). However, this circumstance does not invalidate the proposed procedure and the results obtained from its application.

Both DEM data sets are freely available on the webpage, http://www.ign.es, (accessed on 30 March 2022) of the National Geographic Institute of Spain (IGN), and have the same spatial reference system ETRS89 UTM Zone 30N.

To ensure the overlap of the two grids, and not degrade the quality of the reference (DEM02), the DEM05 data set was interpolated with a $2 \times 2$ mesh step by means of a bilinear interpolation. Following the variance prediction model for the case of bilinear interpolation developed by [4], considering the equality of all the variances of the four positions that intervene in the bilinear interpolation, and the case of a high altimetric correlation; the average variance of the predictor of an altimetric value over any position is equivalent to the variance of the positions involved in the interpolation. In our case, according to the information provided by the metadata, it can be considered to be of the order of 50 cm.

The points analyzed have been obtained through a systematic sampling, for which a grid of 578 rows and 853 columns was generated, which provides a sample size of $n = 493{,}034$. The discrepancies are in the interval $(-54.88, 77.42)$ m; the mean value of the discrepancies is 0.00062 m and the standard deviation 0.41835 m. A general spatial vision of discrepancies appears in Figure 3. Usually, the values assumed for the discrepancies between a product and a reference must be close to zero, but in this case, the above-mentioned observed interval means the presence of extreme values (outliers). Therefore, these data present some extreme points, both on the left and the right. Moreover, the Fisher asymmetry coefficient is $-11.46521$ and the Fisher coefficient of kurtosis is 1009.753; both of them are very high in respect to the normal distribution.

Figure 4 shows the data histogram of the complete data set. Due to the presence of a relatively small number of extreme values, and the histogram showing the distribution concentrated around 0, and due to the effect of the scale of the $x$ axis, the values farthest from 0 are not visible. In order to see the shape of the histogram in more detail, Figure 5 shows the histogram constrained to the interval $(-1, 1)$, which contains 97.69% of observed discrepancies.

Finally, the overall non-normality of discrepancy data may be also observed in Figure 6, where the QQ-plot is shown together with the expected normal line. These graphics suggest a great deviation of expected normality. This situation opens the possibility that the underlying discrepancy data model comes from a finite mixture of normal distributions.

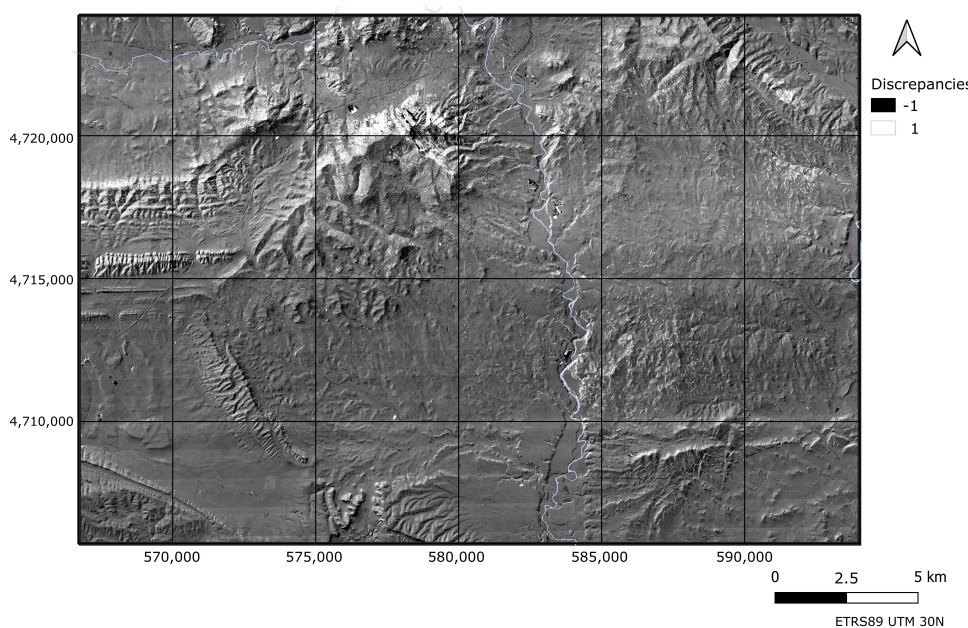

**Figure 3.** Discrepancies model (error model) of the Allo zone (DEM05-DEM02).

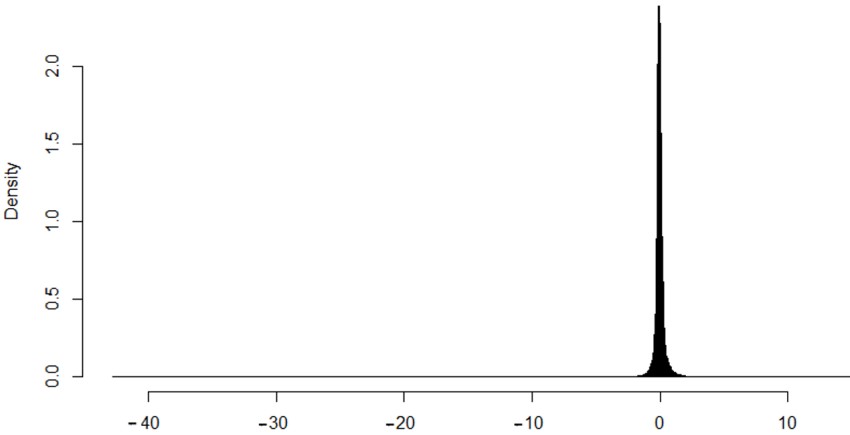

**Figure 4.** Histogram of the discrepancies model (error model) of the Allo zone (DEM05-DEM02).

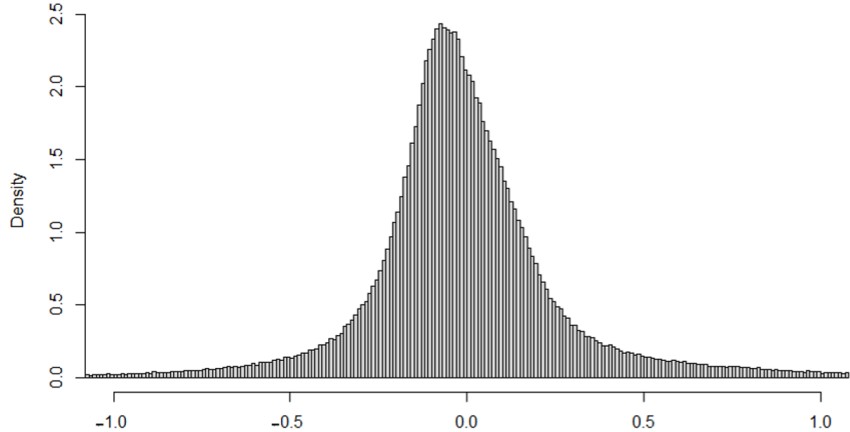

**Figure 5.** Trimmed histogram of the discrepancies model (error model) of the Allo zone (DEM05-DEM02).

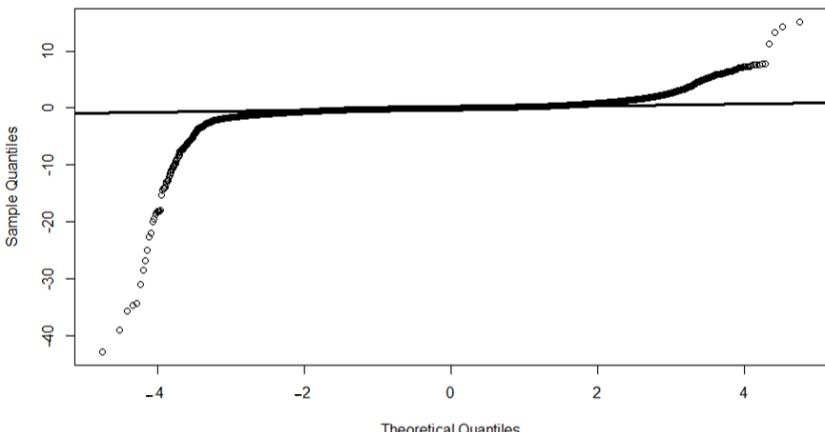

**Figure 6.** Normal QQ-plot of the discrepancies model (error model) of the Allo zone (DEM05-DEM02).

## 5. Results

### 5.1. The Finite Mixture Model

As indicated in Section 3, the decision of the proposed *FMM* is based on the obtained values for *AIC* and *BIC* criteria. Table 1 shows values for both criteria when the number of mixtures, *g*, goes from 2 to 10. Because the estimation procedure is iterative, to show the complexity of the process, the last column includes the number of iterations needed to achieve convergence.

**Table 1.** Values of AIC and BIC for g from 2 to 10 for the estimation of the finite mixture model.

| *g* | *AIC* | *BIC* | Iterations |
|---|---|---|---|
| 2 | 125,846.6 | 125,913.2 | 48 |
| 3 | 81,894.5 | 81,994.5 | 168 |
| 4 | 77,244.1 | 77,377.4 | 1059 |
| 5 | 74,966.8 | 75,133.5 | 1923 |
| 6 | 74,290.3 | 74,490.2 | 4757 |
| 7 | 74,173.8 | 74,407.1 | 13,332 |
| 8 | 74,179.8 | 74,446.4 | 115,186 |
| 9 | 74,166.2 | 74,466.2 | 461,837 |
| 10 | 74,167.9 | 74,501.2 | 400,682 |

In this case, and due to the sampling size being very large, the BIC criterion is adopted [37]. According to Table 1, a mixture of seven normal distributions is proposed. Table 2 shows the vector of estimated parameters, $\hat{\Theta}$, obtained, where $(\hat{\mu}_i, \hat{\sigma}_i)$ are the parameters of the *i*-th normal distribution component and $\hat{\pi}_i$ the probability of this component in the mixture.

**Table 2.** Estimated parameters for each component of the finite mixture model based on 7 normal distributions.

| Component | $\hat{\mu}_i$ | $\hat{\sigma}_i$ | $\hat{\pi}_i$ |
|---|---|---|---|
| 1 | −7.78135 | 10.22195 | 0.00025 |
| 2 | −0.01837 | 0.26977 | 0.18361 |
| 3 | −0.08378 | 0.05688 | 0.08837 |
| 4 | 0.06209 | 0.51793 | 0.16441 |
| 5 | −0.02414 | 0.13835 | 0.52425 |
| 6 | 0.32596 | 0.94185 | 0.03558 |
| 7 | 1.19120 | 2.59239 | 0.00353 |

It can be observed that the first component includes all extreme values on the left, and that the seventh component covers the extreme values on the right. Both cases account for a very low probability. The most important is component five (a half of the population, see $\hat{\pi}_5$).

The estimated population density can be calculated according to (4) and compared with the empirical distribution of the observed data (EDOD) (see Figure 3). A graphical comparison appears in Figure 7, where the EDOD histogram is represented together with the estimated density (the FMM)—the curve in orange. For visibility, the range is trimmed in the interval $[-1, 1]$. The maximum distance detected between both curves is 0.00041, which is a really small value and has a $p$-value greater than 0.1 in the Kolmogorov-Smirnov goodness-of-fit test (the critical value in this case is $1.228/\sqrt{493{,}034} = 0.00175$).

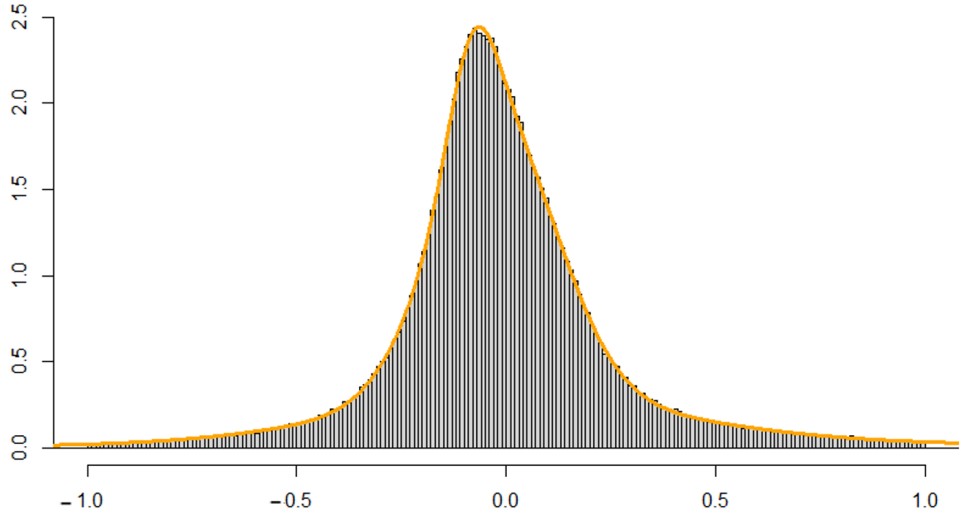

**Figure 7.** Observed histogram (EDOD) and density function derived from the finite mixture model (FMM) for the discrepancies (colored curve).

The *FMM* provides a whole description about the population of discrepancies and allows the calculation of all population parameters and probabilities using Equations (8) and (9). In this case, $\mu_{FMM} = 0.00062$ m, variance $\sigma^2_{FMM} = 0.17502\,\text{m}^2$, and standard deviation $\sigma_{FMM} = 0.41835$ m. Comparing these values derived from the *FMM* model with those corresponding to the *EDOD* (see Section 4), it can be observed that they are the same.

*5.2. Comparison of Approaches*

Because most PAAMs assume a model based on an only one normal distribution component (a 1*NDM*), it is of worth to compare results derived from the proposed *FMM* with the 1*NDM* and the *EDOD*. Results for these models are demonstrated in Table 3, where the similitude can be observed between results provided by the *FMM* model with those of the *EDOD* model, and that the 1*NDM* is very far from them.

With the same idea, and only as an example, Table 4 shows the calculated of probabilities for intervals defined by several values, and compares results obtained using the three models. As occurs on Table 3, it is observed that the 1*NDM* has a bad behavior, whereas the estimated *FMM* fits adequately. In particular, the *FMM* adequately captures both the high concentration of values around the mean, and the tails of the observed data.

**Table 3.** Comparison of quantiles for the Finite Mixture Model (*FMM*), the One Normal Distribution Model (1*NDM*) and the empirical distribution of observed data (*EDOD*).

| Quantile | Value | | |
|---|---|---|---|
| | *EDOD* | *FMM* | 1*NDM* |
| 2.5% | −0.61349 | −0.61378 | −0.82057 |
| 5% | −0.42628 | −0.42648 | −0.68751 |
| 10% | −0.27963 | −0.27943 | −0.53553 |
| 25% | −0.13934 | −0.13953 | −0.28157 |
| 50% | −0.02975 | −0.02980 | 0.00062 |
| 75% | 0.10638 | 0.10620 | 0.28279 |
| 90% | 0.30169 | 0.30120 | 0.53676 |
| 95% | 0.53619 | 0.53678 | 0.68875 |
| 97.5% | 0.81286 | 0.81407 | 0.82057 |

**Table 4.** Example of probabilities for several discrepancy intervals using de Finite Mixture Model (*FMM*), the One Normal Distribution Model (1*NDM*) and the empirical distribution of observed data (*EDOD*).

| Interval (m) | Value | | |
|---|---|---|---|
| | *EDOD* | *FMM* | 1*NDM* |
| $X < -0.5$ | 0.03778 | 0.03767 | 0.11572 |
| $X < -1$ | 0.00705 | 0.00706 | 0.00838 |
| $0.5 < X < 0.8$ | 0.02919 | 0.02927 | 0.00883 |
| $X > 0.5$ | 0.05502 | 0.05513 | 0.11623 |
| $X > 0.41835$ | 0.06890 | 0.06908 | 0.15901 |
| $|X| > 0.01$ | 0.95807 | 0.95781 | 0.98093 |
| $|X| > 0.05$ | 0.78908 | 0.78964 | 0.90487 |
| $|X| > 0.10$ | 0.40906 | 0.40939 | 0.18892 |
| $|X| > 0.20$ | 0.31594 | 0.31600 | 0.63260 |
| $|X| > 0.50$ | 0.09280 | 0.09280 | 0.23202 |
| $|X| > 1$ | 0.02305 | 0.02319 | 0.01683 |

*5.3. Analyzing the Sampling Distributions*

The advantage of an *FMM* is that it allows working with a parametric model that describes the entire discrepancies' population. In order to utilize the model for building a hypothesis test, and in order to accept or reject some assumptions related to the population, it is necessary to know the sampling behavior of the estimators in a sample of size $n$, which is a collection of n independent random variables, all of them distributed according to the distribution of the discrepancy's population. If a 1*NDM* is assumed, the distribution of the mean and variance of the sample are well known. But in this case, we need to know the sampling distribution under the *FMM* obtained. To know this sampling distribution, a simulation procedure was carried out, where 5000 samples for different sampling sizes were obtained. Table 5 shows the values for the mean and standard deviation of each set of 5000 samples. It can be noted that the sampling mean is always a random variable with an expected value that equals to $\mu$ and standard deviation equals to $\sigma / \sqrt{n}$. The third column of Table 5 shows the values of $\hat{\sigma} = s_n \sqrt{n}$, which is very close to the standard deviation of the theoretical model. Additionally, this table shows that the square root of the mean of variances is still a more unbiased estimator for the population standard deviation that the mean of the standard deviation.

**Table 5.** Mean, standard error of the mean, estimated standard deviation of the population, mean of variances, mean of standard deviations and square root of the mean of variances for each simulated sample size *n* (based on 5000 iterations).

| *n* | FMM | | | | | |
| --- | --- | --- | --- | --- | --- | --- |
| | **Mean** | **sd** | $\hat{\sigma}$ | $\bar{s_n^2}$ | $\bar{s_n}$ | $\sqrt{\bar{s_n^2}}$ |
| 20 | 0.00186 | 0.09867 | 0.44126 | 0.2020 | 0.3431 | 0.4494 |
| 30 | 0.00236 | 0.07750 | 0.42446 | 0.1796 | 0.3497 | 0.4238 |
| 40 | 0.00654 | 0.06791 | 0.42952 | 0.1797 | 0.3559 | 0.4240 |
| 50 | 0.00518 | 0.05921 | 0.41868 | 0.1734 | 0.3593 | 0.4164 |
| 80 | 0.00558 | 0.04772 | 0.42678 | 0.1727 | 0.3675 | 0.4156 |
| 100 | 0.00626 | 0.04149 | 0.41488 | 0.1725 | 0.3700 | 0.4154 |
| 200 | 0.00586 | 0.02899 | 0.40995 | 0.1742 | 0.3825 | 0.4173 |
| 500 | 0.00555 | 0.01853 | 0.41428 | 0.1715 | 0.3912 | 0.4141 |

In a statistical hypothesis test, the test statistic is compared with the corresponding quantile of its own sampling distribution under the null hypothesis at the desired confidence level (e.g., $\alpha = 0.05$). For instance, in the normal case when the standard deviation is unknown (as occurs in the EMAS test), the sampling distribution of the test statistic $T = \sqrt{n}(\bar{x} - \mu_0)/s_{n-1}$ is a *t*-Student distribution with $n - 1$ degrees of freedom, which is easily obtained. Something similar occurs in the case of the variance test, where the sampling distribution is a $\chi_{n-1}^2$. Nevertheless, in the case of the application of a *FMM*, these quantiles are not known in advance, but they can be obtained through a simulation process. In our case, by means of the above-mentioned simulation process, we were able to determine the quantiles through the 5000 samples generated to derive Table 5. These quantiles appear in Table 6 for the mean, and in Table 7 for the variance. These tables may be used for finding the critical values in the case of the statistical hypothesis test for the mean and the variance.

**Table 6.** Empirical quantiles in the Finite Mixture Model distribution of means for each sample size (*n*) (based on 5000 simulations).

| *n* | Quantiles | | | | | | | |
| --- | --- | --- | --- | --- | --- | --- | --- | --- |
| | **0.01** | **0.025** | **0.05** | **0.1** | **0.9** | **0.95** | **0.975** | **0.99** |
| 20 | −0.2085 | −0.1410 | −0.1170 | −0.0885 | 0.1040 | 0.1410 | 0.1780 | 0.2240 |
| 30 | −0.1673 | −0.1280 | −0.1017 | −0.0763 | 0.0870 | 0.1170 | 0.1464 | 0.1950 |
| 40 | −0.1445 | −0.1035 | −0.0832 | −0.0650 | 0.0838 | 0.1095 | 0.1370 | 0.1760 |
| 50 | −0.1348 | −0.0946 | −0.0752 | −0.0570 | 0.0724 | 0.0962 | 0.1192 | 0.1510 |
| 80 | −0.1068 | −0.0789 | −0.0630 | −0.0450 | 0.0609 | 0.0814 | 0.0980 | 0.1174 |
| 100 | −0.0988 | −0.0672 | −0.0521 | −0.0400 | 0.0554 | 0.0692 | 0.0839 | 0.1033 |
| 200 | −0.0764 | −0.0501 | −0.0393 | −0.0279 | 0.0403 | 0.0500 | 0.0596 | 0.0729 |
| 500 | −0.0449 | −0.0326 | −0.0249 | −0.0172 | 0.0282 | 0.0348 | 0.0416 | 0.0477 |

*5.4. Application to Postional Accuracy Assessment Methods*

The analyzed PAAMs in this paper are based on the statistical hypothesis test on proportions (NMAS), the mean and deviation (EMAS), but also on the result of estimation processes (NSSDA). These situations are very different, but the *FMM* can be applied to all of them and it is valuable to compare the result of this application with the results of applying the 1*NDM*, which represents the traditional approach. In this subsection, the philosophy of each of these three standards is applied using the *FMM*, and the results are compared with those obtained, assuming the 1*NDM* approach.

**Table 7.** Empirical quantiles in the Finite Mixture Model distribution of variances for each sample size (*n*) (based on 5000 simulations).

| *n* | Quantiles | | | | | | | |
|---|---|---|---|---|---|---|---|---|
| | **0.01** | **0.025** | **0.05** | **0.1** | **0.9** | **0.95** | **0.975** | **0.99** |
| 20 | 0.0183 | 0.0226 | 0.0274 | 0.0341 | 0.2490 | 0.3553 | 0.5785 | 1.2729 |
| 30 | 0.0234 | 0.0288 | 0.0344 | 0.0430 | 0.2424 | 0.3450 | 0.6015 | 1.2833 |
| 40 | 0.0271 | 0.0332 | 0.0393 | 0.0483 | 0.2373 | 0.3796 | 0.6489 | 1.2209 |
| 50 | 0.0332 | 0.0393 | 0.0438 | 0.0527 | 0.2351 | 0.3466 | 0.5653 | 1.2717 |
| 80 | 0.0430 | 0.0488 | 0.0558 | 0.0639 | 0.2354 | 0.3555 | 0.5391 | 0.9611 |
| 100 | 0.0470 | 0.0532 | 0.0593 | 0.0671 | 0.2302 | 0.3282 | 0.5038 | 1.2471 |
| 200 | 0.0624 | 0.0678 | 0.0736 | 0.0811 | 0.2371 | 0.3358 | 0.5814 | 1.5181 |
| 500 | 0.0783 | 0.0829 | 0.0879 | 0.0940 | 0.2250 | 0.3483 | 0.6613 | 1.1634 |

5.4.1. National Map Accuracy Standard

There are several PAAMs based on the proportion test; one of the most popular methods is NMAS (Appendix A), but others exist (e.g., [18,40]). Basically, these methods work by setting a metric tolerance and a maximum case ratio value that cannot exceed the proportion. The control sampling is carried out, the number of observations (discrepancies) that exceed that tolerance is counted, and it is verified that the proportion of cases that exceed the metric tolerance is less than the established proportion. If the observed proportion is greater than the tolerance the product is rejected. The application in this case is immediate. Let $x_H$ be the desired metric tolerance value, and $\pi_H = P[|X| > x_H]$ be calculated in the *FMM* model using Equation (5). Several examples were presented in Table 4, and these probabilities have been used here. The null hypothesis is

$$\mathbb{H}_0 : p \leq \pi_H$$

and the alternative hypothesis:

$$H_1 : p > \pi_H$$

where $p$ is the proportion of sampling discrepancies values that are greater than $x_H$ in a sample of size $n$. Table 8 shows the proportion of times the null hypothesis is rejected (for $\alpha = 0.05$) when M = 5000 samples are taken, and for several metric tolerances (0.01, 0.05, 0.10, 0.15, 0.20 and 0.5) [m] when using the discrepancies between the DEM05 and DEM02. We observe that, in all cases, the test based on the *FMM* performs better than the test based on the 1*NDM*. This means that, for the *FMM*, the rejection value when $\mathbb{H}_0$ is true (type I error) is closer to the desired value (0.05). This does not occur for the lowest tolerance of those considered and when the sample size is small, but it does for the rest of the cases. In the case of the 1NDM, the values are usually less than 5%, which indicates that its statistical behavior is not as expected. This generates uncertainty in its applicability, as it does not generate the level of rejection consigned. It behaves more laxly than expected.

The results in Table 8 clearly indicate that the *FMM* performs statistically much better than the 1*NDM*. Extreme cases are relevant. For very small discrepancy tolerances (0.01 m), and small sample sizes, we observe that the two approaches (*FMM* and 1*NDM*) offer high rejection levels for sample sizes usually recommended in PAAMs (size in the order of 20 elements). However, when the sample size is large (200 or 500), the *FMM* offers rejection values close to the established level of significance. For the tolerances of very large discrepancies (1 m) it happens that the 1*NDM* presents a very high level of rejection. In the cases of intermediate tolerance values, the *FMM* adjusts its rejection level to the value established for significance (0.05), while the 1*NDM* generates practically no rejections. Altogether, this means that the 1*NDM* does not work adequately as a statistical model for this case, generating underestimation and overestimation of the producer's risks (type I error).

**Table 8.** NMAS test: proportion of times where the null hypothesis is rejected for several metric tolerances ($\alpha = 0.05$) when using the Finite Mixture Model (FMM) and the One Normal Distribution Model 1NDM (based on 5000 simulations).

| $n$ | Tol [m] | *FMM* | *1NDM* | Tol [m] | *FMM* | *1NDM* |
|---|---|---|---|---|---|---|
| 20 | | 0.4268 | 0.4268 | | 0.0648 | 0.0000 |
| 30 | | 0.2664 | 0.2664 | | 0.0630 | 0.0000 |
| 40 | | 0.1734 | 0.1734 | | 0.0509 | 0.0000 |
| 50 | 0.01 | 0.1248 | 0.1248 | 0.20 | 0.0741 | 0.0000 |
| 80 | | 0.1368 | 0.0302 | | 0.0685 | 0.0000 |
| 100 | | 0.0720 | 0.0098 | | 0.0731 | 0.0000 |
| 200 | | 0.0658 | 0.0022 | | 0.0564 | 0.0000 |
| 500 | | 0.0640 | 0.0000 | | 0.0551 | 0.0000 |
| 20 | | 0.0495 | 0.0083 | | 0.1092 | 0.0003 |
| 30 | | 0.1007 | 0.0074 | | 0.0552 | 0.0000 |
| 40 | | 0.0513 | 0.0013 | | 0.0774 | 0.0000 |
| 50 | 0.05 | 0.0748 | 0.0003 | 0.50 | 0.0894 | 0.0000 |
| 80 | | 0.0691 | 0.0001 | | 0.0621 | 0.0000 |
| 100 | | 0.0816 | 0.0000 | | 0.0826 | 0.0000 |
| 200 | | 0.0663 | 0.0000 | | 0.0487 | 0.0000 |
| 500 | | 0.0445 | 0.0000 | | 0.0607 | 0.0000 |
| 20 | | 0.0805 | 0.0004 | | 0.1092 | 0.0003 |
| 30 | | 0.0789 | 0.0002 | | 0.1518 | 0.1518 |
| 40 | | 0.0575 | 0.0000 | | 0.0641 | 0.2356 |
| 50 | 0.10 | 0.0749 | 0.0000 | 1.00 | 0.1115 | 0.1115 |
| 80 | | 0.0770 | 0.0000 | | 0.1131 | 0.2840 |
| 100 | | 0.0659 | 0.0000 | | 0.0821 | 0.1997 |
| 200 | | 0.0510 | 0.0000 | | 0.0927 | 0.1740 |
| 500 | | 0.0502 | 0.0000 | | 0.0795 | 0.3716 |

### 5.4.2. Engineering Map Accuracy Standard

The EMAS consist on the realization of two independent statistical hypothesis tests; the first one is for the mean and the second one for the variance (Appendix B). The global null hypothesis is rejected if it is rejected in any of them (test statistics are greater than the corresponding quantile). To compare the $1NDM$ and the $FMM$, $M = 5000$ simulations have been carried out using the discrepancies between the DEM05 and DEM02, and both tests (mean and variance) have been made.

In relation to the mean test, we must consider two situations in relation to the hypothesis; the first one is: $\mathbb{H}_0 : \mu = \mu_H$ and for $\alpha = 0.01, 0.05, 0.1$, and the second one is: $\mathbb{H}_1 : \mu < \mu_H$, whereas for $\alpha = 0.9, 0.95, 0.99$, $\mathbb{H}_1 : \mu > \mu_H$, where $\mu_H$ is the model mean. For the test based on the $FMM$ case, the mean value is compared with the corresponding quantile in Table 6; and for the $1NDM$ case, the usual $t$-Student test has been made. Table 9 shows the proportion of times in which the null hypothesis is rejected, both for the $FMM$ and the $1NDM$.

As in the case of the previous simulations, the results are better the closer they are to the significance values considered (0.01, 0.05, ...). The results presented in this table do not indicate a significant difference between the two methods.

In relation to the variance test, the same simulation procedure has been carried out. The null hypothesis is $\mathbb{H}_0 : \sigma^2 = \sigma_H^2$. Now, for the $FMM$, the test is rejected when the test statistics are less than (for $\alpha = 0.01, 0.05, 0.1$; $\mathbb{H}_1 : \sigma^2 < \sigma_H^2$) or greater than ($\alpha = 0.9, 0.95, 0.99$; $\mathbb{H}_1 : \sigma^2 > \sigma_H^2$) the corresponding value in Table 7, whereas in the $1NDM$ case, the test statistic is $\chi = (n-1)S^2/\sigma_H^2$ and the critical value is obtained using the $\chi^2$ distribution with $(n-1)$ degress of freedom. The result appears on Table 10.

**Table 9.** EMAS test: proportion of times where the null hypothesis is rejected (mean case) when using the FMM and the 1NDM (based on 5000 simulations).

| Model | $n$ | Selected Values of $\alpha$ | | | | | |
|---|---|---|---|---|---|---|---|
| | | **0.01** | **0.05** | **0.1** | **0.9** | **0.95** | **0.99** |
| *FMM* | 20 | 0.0074 | 0.0557 | 0.1120 | 0.0984 | 0.0505 | 0.0131 |
| | 30 | 0.0093 | 0.0487 | 0.1041 | 0.0956 | 0.0487 | 0.0093 |
| | 40 | 0.0108 | 0.0610 | 0.1152 | 0.0783 | 0.0397 | 0.0066 |
| | 50 | 0.0103 | 0.0618 | 0.1207 | 0.0846 | 0.0423 | 0.0086 |
| | 80 | 0.0109 | 0.0575 | 0.1262 | 0.0777 | 0.0347 | 0.0072 |
| | 100 | 0.0111 | 0.0739 | 0.1294 | 0.0737 | 0.0395 | 0.0070 |
| | 200 | 0.0113 | 0.0696 | 0.1387 | 0.0726 | 0.0356 | 0.0064 |
| | 500 | 0.0152 | 0.0762 | 0.1533 | 0.0548 | 0.0236 | 0.0043 |
| *1NDM* | 20 | 0.0167 | 0.0764 | 0.1368 | 0.0745 | 0.0265 | 0.0015 |
| | 30 | 0.0173 | 0.0717 | 0.1305 | 0.0778 | 0.0277 | 0.0023 |
| | 40 | 0.0177 | 0.0706 | 0.1309 | 0.0807 | 0.0297 | 0.0028 |
| | 50 | 0.0175 | 0.0669 | 0.1260 | 0.0794 | 0.0324 | 0.0030 |
| | 80 | 0.0160 | 0.0652 | 0.1192 | 0.0862 | 0.0358 | 0.0039 |
| | 100 | 0.0156 | 0.0638 | 0.1150 | 0.0874 | 0.0361 | 0.0035 |
| | 200 | 0.0130 | 0.0579 | 0.1076 | 0.0960 | 0.0397 | 0.0058 |
| | 500 | 0.0112 | 0.0489 | 0.0987 | 0.1022 | 0.0475 | 0.0072 |

The results of this simulation are clear and obvious for all cases: the results based on the *FMM* mixture model are better than those based on the 1*NDM*, in the sense that the proportion of rejection for $\mathbb{H}_0$ is quite similar to the expected probability in the case of the *FMM* and very different for the 1*NDM*. In this case, the 1*NDM* rejects many cases, which excessively increases the producer's risk.

**Table 10.** EMAS test: proportion of times where the null hypothesis is rejected (variance case) when using the FMM and the 1NDM (based on 5000 simulations).

| Model | $n$ | Selected Values of $\alpha$ | | | | | |
|---|---|---|---|---|---|---|---|
| | | **0.01** | **0.05** | **0.1** | **0.9** | **0.95** | **0.99** |
| *FMM* | 20 | 0.0136 | 0.0553 | 0.1005 | 0.0980 | 0.0543 | 0.0113 |
| | 30 | 0.0104 | 0.0484 | 0.1033 | 0.0976 | 0.0533 | 0.0091 |
| | 40 | 0.0074 | 0.0445 | 0.0959 | 0.0983 | 0.0466 | 0.0087 |
| | 50 | 0.0105 | 0.0437 | 0.0914 | 0.0988 | 0.0499 | 0.0065 |
| | 80 | 0.0099 | 0.0527 | 0.1001 | 0.0987 | 0.0535 | 0.0095 |
| | 100 | 0.0096 | 0.0477 | 0.0935 | 0.1035 | 0.0599 | 0.0070 |
| | 200 | 0.0116 | 0.0524 | 0.1028 | 0.1044 | 0.0498 | 0.0075 |
| | 500 | 0.0095 | 0.0450 | 0.0915 | 0.0989 | 0.0427 | 0.0095 |
| *1NDM* | 20 | 0.3942 | 0.3941 | 0.3901 | 0.3896 | 0.4603 | 0.6090 |
| | 30 | 0.4423 | 0.4448 | 0.4434 | 0.3590 | 0.4275 | 0.5531 |
| | 40 | 0.4772 | 0.4809 | 0.4800 | 0.3376 | 0.3956 | 0.5154 |
| | 50 | 0.5076 | 0.5084 | 0.5055 | 0.3213 | 0.3741 | 0.4905 |
| | 80 | 0.5639 | 0.5631 | 0.5676 | 0.2926 | 0.3404 | 0.4359 |
| | 100 | 0.5875 | 0.5908 | 0.5885 | 0.2853 | 0.3219 | 0.4081 |
| | 200 | 0.6445 | 0.6453 | 0.6432 | 0.2656 | 0.2968 | 0.3523 |
| | 500 | 0.6747 | 0.6713 | 0.6733 | 0.2682 | 0.2871 | 0.3246 |

Finally, the EMAS requires passing the two tests (mean and variance) together (logical AND condition), which means that the EMAS is rejected if one of the two tests is rejected. Although the EMAS is performed according to a 1*NDM* as the underlying distribution, the same philosophy can be applied in the case of the *FMM*. Table 11 shows the proportion of

times than the null hypothesis (that in this case is true) is rejected. A consideration about the EMAS is that it does not contemplate any correction, such as that of Bonferroni, for the fact of combining two independent hypothesis tests simultaneously. Following the suggestion by [41], we introduce this correction. For instance, when the global desired significance is $\alpha = 0.10$, Bonferroni's correction implies, in the case of the bilateral mean test, that the critical value to be considered is $\alpha = 0.025$, and for both sides it is $(\alpha/4)$. Table 11 shows the result for such corrections when applying the *FMM* and the 1*NDM*. It may be observed that in both cases the Bonferroni's correction provides a better result, in the sense of the proportion of times the null hypothesis is rejected being nearer to the desired value ($\alpha$ value). The results of the application of the 1*NDM* are worse than those obtained from the *FMM*.

5.4.3. National Standard for Spatial Data Accuracy

The NSSDA follows a different philosophy than NMAS or EMAS (Appendix C). The NSSDA does not propose a statistical hypothesis test. In this case, the estimation of the value corresponding to the 95% quantile is performed (e.g., 5.25 m at 95% confidence). This result is offered to the interested party (the user), who, based on the estimation, finally has to decide whether or not the data product is suitable for his intended use (fitness for use). Therefore, a value is generated and the user implicitly performs an accept/reject process but not in a statistical acceptation/rejection framework.

**Table 11.** EMAS test: proportion of times of global rejections when using the FMM and the 1NDM with (*) and without (**) applying the Bonferroni's correction (based on 5000 simulations).

| Model | $n$ | Selected Values of $\alpha$ | | | |
|---|---|---|---|---|---|
| | | 0.05 (*) | 0.05 (**) | 0.10 (*) | 0.10 (**) |
| *FMM* | 20 | 0.0829 | 0.0378 | 0.1587 | 0.0844 |
| | 30 | 0.0772 | 0.0325 | 0.1547 | 0.0760 |
| | 40 | 0.0727 | 0.0329 | 0.1609 | 0.0727 |
| | 50 | 0.0806 | 0.0387 | 0.1653 | 0.0796 |
| | 80 | 0.0781 | 0.0370 | 0.1533 | 0.0769 |
| | 100 | 0.0916 | 0.0359 | 0.1786 | 0.0911 |
| | 200 | 0.0791 | 0.0318 | 0.1691 | 0.0800 |
| | 500 | 0.0723 | 0.0328 | 0.1596 | 0.0709 |
| 1*NDM* | 20 | 0.1307 | 0.0928 | 0.1965 | 0.1305 |
| | 30 | 0.1366 | 0.1013 | 0.2011 | 0.1355 |
| | 40 | 0.1399 | 0.1105 | 0.2050 | 0.1406 |
| | 50 | 0.1481 | 0.1143 | 0.2090 | 0.1468 |
| | 80 | 0.1598 | 0.1236 | 0.2153 | 0.1572 |
| | 100 | 0.1641 | 0.1303 | 0.2202 | 0.1613 |
| | 200 | 0.1822 | 0.1531 | 0.2350 | 0.1832 |
| | 500 | 0.1973 | 0.1699 | 0.2466 | 0.1983 |

From a statistical point of view, a key aspect of this standard is the behavior of the quantile estimation in respect to the sampling size, which can vary from the theoretical value. Afterwards, this quantile can be typified in order to compare it with the 1.96 parameter used in the NSSDA as the expansion factor for the 95% confidence interval when a 1*NDM* is assumed. Applying the simulation process described above (Section 4), it is possible to compare the results derived from the three approaches under consideration (Table 12). In this table, the mean of the 97.5 quantiles of each sample and for each value of $n$ is presented (column $\mu_{Q_{0.975}}$). Notice that trend demonstrated by these results is in accordance with those of [41] obtained for the 2D case.

**Table 12.** Mean of the distribution of 97.5% quantile and its typified value for the Finite Mixture Model (*FMM*), the One Normal Distribution Model (1*NDM*) and the empirical distribution of observed data *EDOD* (based on 5000 simulations).

| n | EDOD | | FMM | | 1NDM | |
|---|---|---|---|---|---|---|
| | $\mu_{Q_{0.975}}$ | $Z_{Q_{0.975}}$ | $\mu_{Q_{0.975}}$ | $Z_{Q_{0.975}}$ | $\mu_{Q_{0.975}}$ | $Z_{Q_{0.975}}$ |
| 20 | 0.6627 | 1.5825 | 0.6564 | 1.5676 | 0.6936 | 1.6566 |
| 30 | 0.6940 | 1.6575 | 0.6935 | 1.6564 | 0.7249 | 1.7314 |
| 40 | 0.6901 | 1.6483 | 0.6893 | 1.6461 | 0.7348 | 1.7551 |
| 50 | 0.7155 | 1.7088 | 0.7179 | 1.7146 | 0.7499 | 1.7912 |
| 80 | 0.7405 | 1.7686 | 0.7418 | 1.7717 | 0.7740 | 1.8487 |
| 100 | 0.7586 | 1.8119 | 0.7659 | 1.8294 | 0.7873 | 1.8805 |
| 200 | 0.7781 | 1.8585 | 0.7874 | 1.8807 | 0.8010 | 1.9131 |
| 500 | 0.8009 | 1.9129 | 0.8062 | 1.9257 | 0.8121 | 1.9397 |

In order to have a reference for comparing the results of Table 12, the asymptotic case ($n \rightarrow \infty$) is used. For instance, to obtain the constant that multiplies the value of $MSE_z$ in the case of the 1*NDM*, the value corresponding to the 97.5% quantile is 1.96, which is derived as follows:

$$K_{97.5(1NDM)} = \frac{Q_{97.5(1NDM)} - \mu_{1NDM}}{\sigma_{1NDM}} = \frac{0.82057 - 0.00062}{0.41835} = 1.960 \tag{11}$$

The same computation for the *FMM* results:

$$K_{97.5(FMM)} = \frac{Q_{97.5(FMM)} - \mu_{FMM}}{\sigma_{FMM}} = \frac{0.81407 - 0.00062}{0.41835} = 1.944 \tag{12}$$

In consequence, when $n \rightarrow \infty$, in the proposed version of the NSSDA based on the *FMM*, the 1.96 values are replaced by 1.944. This implies that the limit value for $MSE_z$ shall be slightly less than that obtained for the 1*NDM* case. Note that when applying this calculus procedure, we can propose the NSSDA standard for different quantiles, not only the 97.5 that is used by the rule. In following with the results of the simulations, the columns $Z_{Q_{0.975}}$ in Table 12 show the same for all sample size cases and approaches: the typified values obtained by the simulation process are less than the corresponding value for the population ($n \rightarrow \infty$). This means the presence of underestimation of this parameter, in accordance with previous results [42]. This underestimation of the expansion factor leads to underestimating the value corresponding to 95, that is, the sample results in a lower level of positional error than actually exists in the population, which is a risk for the user. The difference of these values with respect to the theoretical ones ($n \rightarrow \infty$,) has the same range of magnitude for the three models (*EDOD*, *FMM* and 1*NDM*). However, the 1*NDM* presents less discrepancy for small sample sizes, and the *EDOD* and *FMM* present less discrepancy for the larger sample sizes.

## 6. Discussion

In relation to the *FMM*, we can highlight that they are a fully developed and applied statistical tools in other fields; however, we do not have knowledge of their application to the case of spatial data, and even less on the subject of positional accuracy. The application of *FMM* is not complex, as has been evidenced in the work; in addition, to show a simpler case we have only worked in 1D (elevation discrepancies). However, the model is directly applicable to 2D and 3D cases if the coordinates and their associated errors are considered independently. Since the tools to fit the model exist, and the selection criteria are common (e.g., AIC, BIC), the most critical aspect is the sample size to make a good fit. This size will depend a lot on the data to be adjusted (informational structure); thus, there is no possibility of offering quantitative recommendations. Obviously, the bigger the sampling

size is, more accurate the estimation is, especially if the hypothesis of a mixture is true. As a first idea, the sample size should be as big as possible, but an important limitation is given by the obtention cost of the sample.

In any case, it is best to proceed with empirical testing; for instance, by some simulation procedures, we found that sample sizes greater than 2000 produce acceptable results regarding the distance between the obtained model (the fitted $FMM$) and the real data (the $EDOD$). An interesting aspect that has not been explored in this work is that once the MMF has been obtained, its results may have other applications. For example, through the estimated model, a grouping can be provided, which is intrinsic to the data and that, unlike the cluster analysis, does not need additional explanatory variables, since it is produced by the ascription of each discrepancy case to that one mixing distribution to which it is most likely to belong. These groups can also try to be interpreted using multivariate statistical techniques such as discriminant analysis, logistic regression, etc. In addition, if other variables are available (e.g., slope, aspect, type of terrain and so on), this situation can help to better understand the nature of the mixing distributions (see, for instance, [31,32,43–45]). The BIC criterion has led us to select a model with seven components. This model offers a majority component (fifth component with 52% of the weight), three components with weights between 5% and 20% and other very minor components, two of them linked to extreme values (atypical/outlier values in the $1NDM$ case). We really do not know if a model with fewer components would work pretty much the same as this seven component model; however, this is not really a problem, because once it is decided to use an $FMM$ type adjustment, its dimension (number of components) is easily managed by means of any statistical tool. For this reason, we consider that the following selection criteria based on BIC offers the same solution, impartial and objective, to anyone who performs the same process on the same data, which allows the method to be standardized.

In relation to the discrepancy data values used in this paper, the analysis carried out comparing the results of the $FMM$ with the $1NDM$ and the observed data ($EDOD$) clearly show that the $FMM$ offers much more consistent results with the real population than the $1NDM$. Thus, the difference in values between the $EDOD$ and the MMF is very small in all the cases presented in Tables 3, 4 and 12. Moreover, if the $1NDM$ is compared with the MMF and the $EDOD$, it can be observed that the difference in quantile distance has reached 23% in the case of 90% quantile (Table 3). In the case of probabilities (Table 4), the probability difference between $1NDM$, the MMF and the $EDOD$ in the analyzed intervals has reached 0.3 (case $X > 0.2$), which means 30% of discrepancy. The above two examples are cases of maximum difference, but on average, the difference is also quite a lot. This clearly demonstrates that the $1NDM$ model is not suitable for modeling data such as those used in this paper.

Finally, we will pay attention to the results when considering commonly used standards for positional accuracy assessment. In this case, the most important thing is the adjustment to the level of significance, as it is the risk of the producer that is assumed in a statistical process of control. As shown by Table 8 for the NMAS, the $FMM$ performs statistically much better than the $1NDM$ when considering all the tolerance values and sample sizes used in the analysis. In the case of the $1NDM$, the values are usually less than 5%, which indicates that its statistical behavior is not "as expected". Table 11 presents the main results for the case of the EMAS. The first conclusion is the need of a Bonferroni correction when applying the EMAS. For both significance levels (0.05 and 0.1), the rejection level by the $FMM$ is a little less than the prescribed level; the differences are in the order [2.1, 1.5]% (always less). The contrary occurs for the $1NDM$; the differences are in the order [5.6, 7.4]% (always more). We consider that these differences with respect to the consigned value are really high. In this case, there exists an excess of rejection that harms the producer, with the consequent problems that this can also generate for the user. The NSSDA is not a statistical test, although it can be understood that it considers a process of acceptance/rejection by the user, as the latter must ask himself whether the result of the estimation seems adequate or not for his application. If we consider that this process is based on the simple comparison

of values (estimated by the sample versus the theoretical), Table 12 indicates the acceptance for all sample sizes and models, and a very similar behavior of the three approaches is under consideration.

## 7. Conclusions

We consider that statistical models based on finite mixtures of normal distributions allow a better approximation to actual altimetric errors, as shown by its ability to fit the observed data. The method and the tools for the application of this alternative are already developed, and its application is quite direct. The main limitation of the use of *FMM*s is the need for large sample sizes to fit the parameters of the mixing distributions. Furthermore, no simple rule can be offered to establish this size. For the application phase of the FMMs using PAAMs, larger sample sizes will be needed, but, in any case, in the order of the previous recommendations for these standards.

The use of the *FMM*s as the statistical models for the application of the PAAMs analyzed (NMAS, EMAS and NSSDA), generates improvements in the behavior of the results for those standards based on statistical hypotheses tests (e.g., NMAS and EMAS). In this case, the *FMM*s application offers results with a better approximation to the levels of significance. If the PAAM is not based on a statistical process, as it is here analyzed for the NSSDA, it does not have such a clear advantage.

Since FMM is a statistical model obtained from the numerical values of the errors, it does not necessarily have to be associated, a priori, with an underlying physical model of the soil. Therefore it can be considered as a black box system, which is common for PAAMs of this type. However, a posteriori, the FMM could be used to analyze the spatial distribution of the mixing distributions in order to get a more ground-based interpretation of the error distribution and the reason of its allocation to each component of the FMM. We believe that this could be of great interest if some relationship is achieved with variables that have traditionally been considered to explain the altimetry error (e.g., slope, vegetation cover). We consider this to be a future line of research that could help establish the use of FMMs for DEM error assessment and analysis.

In this paper, the application has been developed for the case of 1D errors, and for this reason we worked with DEMs, but the method is directly applicable to the case of 2D errors, if the X and Y components are considered independently. Let us bear in mind that the proposed method provides a parametric statistical model, which, once estimated, allows us to work through population values. Therefore, its use is not limited to the case of altimetry errors, which is what has been developed here; it is also useful for obtaining probabilistic models in any set of quantitative measurements, such as slopes or the values of heights themselves. This would allow them to be used, for example, to compare between different areas, or even in the same area in different periods of time. Likewise, knowledge of the theoretical model allows its use when proposing more precise and exact contrasts appropriate to the nature of the data.

**Author Contributions:** Data curation, J.R.-A. and F.J.A.-L.; Formal analysis, J.R.-A. and F.J.A.-L.; Funding acquisition, F.J.A.-L.; Investigation, J.R.-A. and F.J.A.-L.; Methodology, J.R.-A. and F.J.A.-L.; Project administration, F.J.A.-L.; Software, J.R.-A.; Writing–original draft, J.R.-A. and F.J.A.-L.; Writing—review & editing, J.R.-A. All authors have read and agreed to the published version of the manuscript.

**Funding:** This work has been partially financed by the research project "Functional Quality of Digital Elevation Models in Engineering" of the State Agency Research of Spain. PID2019-106195RB-I00 /AEI/10.13039/501100011033 (https://coello.ujaen.es/investigacion/web_giic/funquality4dem/) (accessed on 30 March 2022).

**Institutional Review Board Statement:** Not applicable.

**Informed Consent Statement:** Not applicable.

**Data Availability Statement:** Data sets are freely available on the webpage, http://www.ign.es, accessed on 30 March 2022.

**Conflicts of Interest:** The authors declare no conflict of interest. The funders had no role in the design of the study; in the collection, analyses, or interpretation of data; in the writing of the manuscript, or in the decision to publish the results.

## Appendix A. NMAS

1. Select a sample.
2. Calculate the error of each point in each component:.

$$e_{x_i} = x_{p_i} - x_i, \qquad e_{y_i} = y_{p_i} - y_i, \qquad e_{z_i} = z_{p_i} - z_i$$

   where:

   - $x_i$, $y_i$, $z_i$ are the coordinates in the reference (RDS).
   - $x_{p_i}$, $y_{p_i}$, $z_{p_i}$ are the coordinates in the product (ADS).

3. Calculate the horizontal component of the errors in $x$, $y$ at each point:

$$e_{H_i} = \sqrt{e_{x_i}^2 + e_{y_i}^2}$$

4. Establish which are the maximum tolerable errors:

   - Horizontal: HTol1 = 0.085 cm (1/30 inch) in maps of a scale greater than E20K or HTol2 = 0.05 cm (1/50 inch) in maps at a scale smaller or equal to E20K.
   - Vertical: Half of the equidistance (interval) between contour lines (VTol).

5. Count how many points have a horizontal error $e_H$ greater than the tolerance that applies to the scale case. The control is surpassed in the horizontal component if the number of points having an error above the tolerance does not exceed 10% of the cases.

6. Count how many points have a vertical error $e_z$ greater than the vertical tolerance. The control is surpassed in the vertical component if the number of points that have an error above the tolerance does not exceed 10% of the cases.

## Appendix B. EMAS

1. Select a sample of $n$ points, where $n \geq 20$.
2. Calculate the error for each point in each component:

$$e_{x_i} = x_{p_i} - x_i, \qquad e_{y_i} = y_{p_i} - y_i, \qquad e_{z_i} = z_{p_i} - z_i$$

   where:

   - $x_i$, $y_i$, $z_i$ are the coordinates in the reference (RDS).
   - $x_{p_i}$, $y_{p_i}$, $z_{p_i}$ are the coordinates in the product (ADS).

3. Calculate the mean error of each component:

$$\bar{e}_x = \frac{1}{n} \sum_{i=1}^{n} e_{x_i}; \qquad \bar{e}_y = \frac{1}{n} \sum_{i=1}^{n} e_{y_i}; \qquad \bar{e}_z = \frac{1}{n} \sum_{i=1}^{n} e_{z_i}$$

4. Calculate the sampling standard deviation in each component:

$$S_x = \sqrt{\frac{(e_{x_i} - \bar{e}_x)^2}{n-1}}; \qquad S_y = \sqrt{\frac{(e_{y_i} - \bar{e}_y)^2}{n-1}}; \qquad S_z = \sqrt{\frac{(e_{z_i} - \bar{e}_z)^2}{n-1}}$$

5. Perform, for each component, the standard compliance test to determine whether the mean error is acceptable (which implies an absence of bias). For this, a test is

performed on the mean, under the assumption of unknown population variance and establishing the following hypotheses:

$$\mathbb{H}_0 : \mu = 0; \qquad \mathbb{H}_1 : \mu \neq 0$$

The map will pass the test with a significance level $\alpha$ if the following is met:

$$|t_x| \leq t_{n-1,\alpha/2}; \qquad |t_y| \leq t_{n-1,\alpha/2}; \qquad |t_z| \leq t_{n-1,\alpha/2}$$

where:

- $t_{n-1,\alpha/2}$ Student's $t$-distribution value, with $n-1$ degrees of freedom.
- $t_x, t_y, t_z$: Result of calculating the following statistics:

$$t_x = \frac{\sqrt{n}\bar{e}_x}{S_x}; \qquad t_y = \frac{\sqrt{n}\bar{e}_y}{S_y}; \qquad t_z = \frac{\sqrt{n}\bar{e}_z}{S_z}$$

6. Perform, for each component, the standard compliance test to determine if the sample standard deviation is within acceptable limits. For this purpose, a test is performed on the variance, establishing the following hypotheses in relation to a maximum variance value $\sigma_{0x}^2$, $\sigma_{0y}^2$ and $\sigma_{0z}^2$ pre-established and specified on each component:

$$\mathbb{H}_0 : \sigma^2 \leq \sigma_0^2; \qquad \mathbb{H}_1 : \sigma^2 > \sigma_0^2$$

The product will pass the control with a significance level $\alpha$ if the following is met:

$$\chi_x^2 \leq \chi_{n-1,1-\alpha}^2; \qquad \chi_y^2 \leq \chi_{n-1,1-\alpha}^2; \qquad \chi_z^2 \leq \chi_{n-1,1-\alpha}^2$$

where:

- $\chi_{n-1,1-\alpha}^2$ Theoretical value of the Chi square distribution, with $n-1$ degrees of freedom.
- $\chi_x^2, \chi_y^2, \chi_z^2$: Result of calculating the following statistics:

$$\chi_x^2 = \frac{(n-1)S_x^2}{\sigma_{0x}^2}; \qquad \chi_y^2 = \frac{(n-1)S_y^2}{\sigma_{0y}^2}; \qquad \chi_z^2 = \frac{(n-1)S_z^2}{\sigma_{0z}^2}$$

**Appendix C. NSSDA**

1. Select a sample of $n$ points, where $n \geq 20$.
2. Calculate the error for each point in each component:

$$e_{x_i} = x_{p_i} - x_i, \qquad e_{y_i} = y_{p_i} - y_i, \qquad e_{z_i} = z_{p_i} - z_i$$

where:

- $x_i, y_i, z_i$ are the coordinates in the reference (RDS).
- $x_{p_i}, y_{p_i}, z_{p_i}$ are the coordinates in the product (ADS).

3. Calculate the mean error of each component:

$$MSE_x = \sqrt{\frac{\sum e_{x_i}^2}{n}}; \qquad MSE_y = \sqrt{\frac{\sum e_{y_i}^2}{n}}; \qquad MSE_z = \sqrt{\frac{\sum e_{z_i}^2}{n}}$$

4. Obtain the horizontal $NSSDA_H$ value:

- if $MSE_x = MSE_y$,

$$NSSDA_H = \frac{2.4477}{\sqrt{2}}MSE_r = 2.4477MSE_r$$

where:

$$MSE_r = \sqrt{MSE_x^2 + MSE_y^2}$$

- if $MSE_x \neq MSE_y$ and $0.6 < MSE_{min}/MSE_{max} < 1.0$

$$NSSDA_H = 2.4477 \times 0.5 \times (MSE_x + MSE_y)$$

5. Obtain the vertical $NSSDA_z$ value according to the following expression:

$$NSSDA_z = 1.9600 \times MSE_z$$

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
