# Peer review of "Finite Mixture Models in the Evaluation of Positional Accuracy of Geospatial Data"

_remotesensing, doi:10.3390/rs14092062_

Round 1
Reviewer 1 Report
The authors have attended to the considerations. Thus, I recommend the acceptance of the article for publication.
Author Response
Thank you very much for your kind comment regarding our work.
Reviewer 2 Report
Dear Authors,
Thank you for the response and questions. I apologize for the insufficient information provided in the review reports.
- As a civil engineer, we traditionally learnt and taught the vertical accuracy for topographic mapping has two components, the method and the slope. Consequently, the specification could be presented in the following form.
(1)
A sample specification is then,
|
Terrain slope |
First order |
Second order |
Third order |
|
Less than 2 degree |
0.5 |
0.7 |
1.0 |
|
2- 6 degree |
1.2 |
1.7 |
2.5 |
|
6-25 degree |
2.5 |
3.3 |
5.0 |
|
Above 25 degree |
5.0 |
6.7 |
10.0 |
Yes, this index is designed with an error model, the traditional one. Overlooking the slope effect, the data would not fulfill the single distribution assumption.
This concept originated quite long ago. A well-known form is shown as the equation (8.7) in Li, et al. (2004).
(2)
The authors may find this equation in other literatures and cite those instead. For example, Koelbl (2001) reviewed several national specifications of European countries.
Koelbl O., (2001): Technical Specifications for the Elaboration of Digital Elevation Models. EPFL.
- When the airborne lidar has been widely applied, another factor, the vegetation coverage, is considered important. The form of equation becomes,
(3)
Where t is the average vegetation height. If airborne lidar is used, overlooking this factor would result either biased or multi-modal distribution as well.
This equation is referenced to an engineering specification dated 2004 and has been used by this reviewer and his colleagues since. There were comprehensive studies made for this model in the engineering report, but to the best knowledge of this reviewer, this is not published in a journal article. The reason is that this type of work is generally considered as technical, not scientific. Prof. Z.L. Li and others may have performed similar studies and published in journal. A literature survey would be required.
- This reviewer agrees that the simplified evaluation scheme adopted by USGS is not “rigorous”, and, welcome a discussion from the finite mixture model view, even though this model would not be able to provide a practical solution. Most importantly, based the discussion this reviewer had with USGS staff members, the adoption of a simple index is for the ease of operation. The USGS staffs are fully understand the physical factors behind DEM uncertainties. Secondly, unless there is a large portion of the terrain with a specific slope, the distribution will not be multi-modal. That is, whether the basic assumption to justify finite mixture model is satisfied, would be case dependent.
The above descriptions are stated as the reference. Based on these, this reviewer would humbly suggest,
- Find out the physical explanation of the result obtained with the finite mixture model in the current study, by relating to the fundamental DEM error model.
- Although the data used is acquired with airborne lidar, the analysis may be limited to the traditional index, equation 2, to lessen the effort. One would need the vegetation layer and additional Digital Canopy Model/Building Model for performing analysis of equation 3. These layers are normally documented in the LAS file. However, this reviewer does not sure whether the authors have these layers. Meanwhile, this reviewer could not provide a well-known literature for equation 3 without further literature search.
- As the most convenient way for the revision, the authors may just include a paragraph for explaining the likely cause of “mixtures”, and discussing the possible limitation of the proposed “statistical analysis” method which ignores the physical factors of the errors.
Best regards.

Author Response
Attached we send the point-by-point response to the referee.

Reviewer 3 Report
In my opinion it is a well prepared manuscript in terms of technical and content. Well-chosen research methods, appropriate research area, The authors presented the interesting research about replace the univariate normal statistical model by the FMM model.
I think that the presented topic is original and brings new insights in the evaluation of positional accuracy of geospatial data.
The current manuscript is better prepared than the original (previous version). The most important (positive) changes include:
• The title of the works was corrected and the scope and purpose of the research were clarified,
• The scope of research and a review of the literature on the subject were extended
• The figures have been improved - they are now of much better quality
• The structure of the manuscript has been improved, improving its clarity and readability.
• Most of the reviewers' comments were taken into account (or justification was provided).
The methodology, course of work, their analysis, and interpretation have been properly planned, carried out and described in the text. The results were correctly presented and relate to the issues discussed in the manuscript. All references cited have been properly prepared and used. The introduction to the topic has been properly and meticulously prepared.
My only remark relates to the summaries and conclusions. Please, discuss and present them in greater detail, especially in terms of applicability of your solutions (based on the results of your work).
Some fragments of text (i.n. lines 109-118) require stylistic editing.
Author Response
In my opinion it is a well prepared manuscript in terms of technical and content. Well-chosen research methods, appropriate research area, The authors presented the interesting research about replace the univariate normal statistical model by the FMM model.
I think that the presented topic is original and brings new insights in the evaluation of positional accuracy of geospatial data.
The current manuscript is better prepared than the original (previous version). The most important (positive) changes include:
• The title of the works was corrected and the scope and purpose of the research were clarified,
• The scope of research and a review of the literature on the subject were extended
• The figures have been improved - they are now of much better quality
• The structure of the manuscript has been improved, improving its clarity and readability.
• Most of the reviewers' comments were taken into account (or justification was provided).
The methodology, course of work, their analysis, and interpretation have been properly planned, carried out and described in the text. The results were correctly presented and relate to the issues discussed in the manuscript. All references cited have been properly prepared and used. The introduction to the topic has been properly and meticulously prepared.
thank you very much for your kind consideration to our work
My only remark relates to the summaries and conclusions. Please, discuss and present them in greater detail, especially in terms of applicability of your solutions (based on the results of your work).
Some considerations have been added in the Conclusion section that, we hope, could clarify these aspects.
Some fragments of text (i.n. lines 109-118) require stylistic editing.
This text has been rewritten in this sense
This manuscript is a resubmission of an earlier submission. The following is a list of the peer review reports and author responses from that submission.
Round 1
Reviewer 1 Report
General comments
The authors propose the application of the Gaussian Finite Mixture Model (FMM) in the analysis of positional discrepancies in the spatial data quality process. Specifically, the model is applied to altimetric data from digital elevation model (DEM). In addition, the authors present the sampling behavior of the estimators (mean and standard deviation) under this model, as well as FMM application in the methodology of NMAS, EMAS and NSSDA standards. The results obtained show the efficiency of the FMM.
The article is relevant and provides a scientific gain by analyzing the positional quality of geospatial data using FMM. This process is an interesting option to work with non-normal data, being a subject of interest to the scientific community focused on spatial data quality.
In general, the text is clear and structured, needing more attention to topics 3 and 4.
The problem and the hypothesis of the research are well defined.
The bibliographical references need attention. Only 10 of the 40 references cited were published in the last 5 years.
The methodology and experiment are appropriate to achieve the objectives. However, some questions are presented in the specific comments, with the focus of clarifying the methodology used.
The discussion of the results is clear and consistent.
Specific Comments
>> Large and very specific title. Suggestion: Finite mixture models in the evaluation of positional accuracy of geospatial data
>> The abstract starts with the term "Digital terrain models (DEM)". This term is also presented in the text of the article. The acronym for "Digital Elevation Model" is DEM and for "Digital terrain model" is DTM. Although there is no standard in the definitions, DTM and DEM are different concepts. I suggest standardizing on the text the term "Digital Elevation model (DEM)".
>> Page 2 - lines 40-41: the term "5.27 m ± 0.15 m" represents a value and precision, not a confidence interval.
>> Page 2 - lines 47-48: the term "complete and modern" seems self-promoting.
>> Page 4 - lines 144-145: the authors say that they used all data discrepancies (exposed in section 4) to estimate the FMM. However, in section 4 it presents n=493034. Is "n" the total discrepancies in the study area? How was the discrepancy sample obtained? Was it a subtraction between the DEMs, pixel by pixel? For the area of 500km², a DEM of 5m resolution would result in 20,000,000 discrepancies.
>> Page 4 - lines 144-145: Was the spatial distribution of the checkpoints used as a parameter for sample representativeness?
>> Page 5 - line 157: It should have an explanation and description the first time it introduces the acronym "1NDM".
>> Page 5 - line 159: what are these parameters?
>> Figure 1: confusing the flowchart, mainly in the description of the process for each approach.
>> In general, section 3 of the article is confusing.
>> Section 4 - Paragraph 1: A figure of the study area is essential.
>> Page 5-6 - lines 183-192: In the description of each DEM two RMSE values are presented. The reader will get confused.
>> Page 6 - lines 196-197: Resampling the DEM will change the pixel altitudes values, consequently changing the image statistics. I don't see how I can best use this procedure. Could you have another approach?
>> Page 6 - line 204: How were the discrepancies obtained? Was it by subtraction of DEMs or by homologous points? If it was by subtraction of the DEMs, did you not consider the horizontal uncertainty between the DEMs? If you disregarded the horizontal uncertainty of the DEMs, what is the influence on the results? Could the FMM modeling be different or inefficient?
>> Page 6 - Figure 2: This figure is poor in cartographic information and does not achieve the goal of adding knowledge to the article. By the description of the figure, I suppose that it is the result of the subtraction of the DEMs. It lacks the scale of values of the theme presented, coordinate grid, location map.
>> Page 7 - Figure 3: It is essential to show the histogram in full. With all the data one can analyze various items, e.g. outliers. Title and coordinate axis show the term "datos". Use the term in English, not Spanish
>> Since there is topic 6 "discussion", I suggest that topic 5 be just "Results".
>> Page 8 - line 224: cite references to say that the use of "BIC" is indicated for very large samples.
>> Page 8 - Figure 5: it is necessary to present the whole curve. Presented only the interval -1 to +1.
>> Page 9 - Table 3: To be symmetric the quantile values should have the 2.5% quantile.
>> Page 11 - First paragraph of topic 5.4 does not have the lines numbered.
>> Page 11 - 1st paragraph: "NMAS Standard" should be a subtopic (e.g. 5.4.1)
>> Page 12 - Topic "5.5. EMAS Standard" should be subtopic 5.4.2
>> Page 12 - lines 313-314: The authors formulate the test hypotheses for means and variances with two H1 (major and minor). Would it not be more appropriate to use a two-sided test (with H1: μ ≠ μH), as in Appendix B ?
>> Page 15 - Topic "5.6. NSSDA standard" should be subtopic 5.4.3
>> Page 16 - line 400: lacks bibliographic references to cite the suggested cluster analysis from the FMM.
Author Response
Dear reviewer, thank you very much for your comments and suggestions.
Q<<The bibliographical references need attention. Only 10 of the 40 references cited were published in the last 5 years.>>
A: We have added more updated references, related to several aspects on the document
Q<< Large and very specific title. Suggestion: Finite mixture models in the evaluation of positional accuracy of geospatial data >>
Ok. Done. Thanks for your suggestion.
Q<<The abstract starts with the term "Digital terrain models (DEM)". This term is also presented in the text of the article. The acronym for "Digital Elevation Model" is DEM and for "Digital terrain model" is DTM. Although there is no standard in the definitions, DTM and DEM are different concepts. I suggest standardizing on the text the term "Digital Elevation model (DEM)".>>
A: Corrected.
Q<<Page 2 - lines 40-41: the term "5.27 m ± 0.15 m" represents a value and precision, not a confidence interval. >>
A: It is an example. In this case, 0.15 is the amplitude of the IC, not only the standard error.
Q<<Page 2 - lines 47-48: the term "complete and modern" seems self-promoting.>>
A: This phrase has been changed
Q<<Page 4 - lines 144-145: the authors say that they used all data discrepancies (exposed in section 4) to estimate the FMM. However, in section 4 it presents n=493034. Is "n" the total discrepancies in the study area? How was the discrepancy sample obtained? Was it a subtraction between the DEMs, pixel by pixel? For the area of 500km², a DEM of 5m resolution would result in 20,000,000 discrepancies. >>
Q<<Page 4 - lines 144-145: Was the spatial distribution of the checkpoints used as a parameter for sample representativeness?>>
The way of obtaining the sample has been specified in the article, and it consists of a systematic sampling in a grid of 578 by 853 elements so that one point has been obtained for each element. That provides the 493034 points studied
Q<<Page 5 - line 157: It should have an explanation and description the first time it introduces the acronym "1NDM". >>
A: Explained
Q<<Page 5 - line 159: what are these parameters? >>
A: Parameters of the finite mixture model. They appear on Table 2. In this part we explain the process of analysis that we have applied later. To clarify this question, we have rewritten the sentence.
Q:<<Figure 1: confusing the flowchart, mainly in the description of the process for each approach.
A: The figure has been improved in order to present the process properly.
Q:<<In general, section 3 of the article is confusing. >>
A: A rewording of several sentences has been done to clarify the process.
Q<< Section 4 - Paragraph 1: A figure of the study area is essential.>>
A: A figure has been added
Q: <<Page 5-6 - lines 183-192: In the description of each DEM two RMSE values are presented. The reader will get confused. >>
A: The RMSE of the LiDAR source has been eliminated.
Q: <<Page 6 - lines 196-197: Resampling the DEM will change the pixel altitudes values, consequently changing the image statistics. I don't see how I can best use this procedure. Could you have another approach? >>
A: You are right, but as indicated later, the level of uncertainty introduced by this operation can be known (see the indicated reference), which is in the order of the uncertainty of the interpolated data.
Q: <<Page 6 - line 204: How were the discrepancies obtained? Was it by subtraction of DEMs or by homologous points? If it was by subtraction of the DEMs, did you not consider the horizontal uncertainty between the DEMs? If you disregarded the horizontal uncertainty of the DEMs, what is the influence on the results? Could the FMM modeling be different or inefficient?
A: The executed process is a subtraction between the elevations of the two models.
Of course, there is a positional uncertainty. However, it has not been considered.
When working with two DEM grids it is usual to subtract them.
The best way to avoid consideration of horizontal positional uncertainty is to have a true reference of greater accuracy than the product being evaluated. However, this is not the case, so a clarifying comment has been added.
On the other hand, the consideration of positional uncertainty would lead us to introduce more models and assumptions, which may not be clarifying. In this case, the horizontal positional uncertainty of the two models is of the same order, as indicated by the producer's metadata, we consider that this leads to a compensated situation.
Q<<Page 6 - Figure 2: This figure is poor in cartographic information and does not achieve the goal of adding knowledge to the article. By the description of the figure, I suppose that it is the result of the subtraction of the DEMs. It lacks the scale of values of the theme presented, coordinate grid, location map.>>
A: A new figure has been added
Q:<<Page 7 - Figure 3: It is essential to show the histogram in full. With all the data one can analyze various items, e.g. outliers. Title and coordinate axis show the term "datos". Use the term in English, not Spanish. >>
A: We have added it.
However, for the sake of visibility, the trimmed histogram also appears on the paper.
Q:<<Since there is topic 6 "discussion", I suggest that topic 5 be just "Results".
A: Done
Q: <<Page 8 - line 224: cite references to say that the use of "BIC" is indicated for very large samples. >>
A: References are 33 and 34, Cameron, A. C., Trivedi, P. K. and Burnham, K. P., Anderson, D. R., 2003. They are referred to in section 2, and for this reason, they did not appear here. This comment also appears on lines 124-126. Nevertheless, a reference has been added.
Q:<<Page 8 - Figure 5: it is necessary to present the whole curve. Presented only the interval -1 to +1. >>
A: We have added the whole histogram, that presents a very sharped shape. In this case, the whole histogram difficult to see the fit between the orange line (theoretical model) and the histogram (observed data). For this reason, it has not been added.
Q:<<Page 9 - Table 3: To be symmetric the quantile values should have the 2.5% quantile. <<
A: Done
Q:<<Page 11 - First paragraph of topic 5.4 does not have the lines numbered.
A: Corrected
Q:<<Page 11 - 1st paragraph: "NMAS Standard" should be a subtopic (e.g. 5.4.1)
A: Done
Q: <<Page 12 - Topic "5.5. EMAS Standard" should be subtopic 5.4.2.
A: Done
Q:<<Page 12 - lines 313-314: The authors formulate the test hypotheses for means and variances with two H1 (major and minor). Would it not be more appropriate to use a two-sided test (with H1: μ ≠ μH), as in Appendix B?
A: In this case we only provide a tail because of the absence of symmetry in the FMM and for make easier the comparation between models. For this reason, the table is made for the unilateral case. But, Table 11 shows the true EMAS test, using the bilateral case for the mean and the unilateral case for the variance
Q:<<Page 15 - Topic "5.6. NSSDA standard" should be subtopic 5.4.3
A: Done
Q:<<Page 16 - line 400: lacks bibliographic references to cite the suggested cluster analysis from the FMM.
A: Done

Reviewer 2 Report
This article reported an exercise with an existing well developed statistical method targeted for DEM accuracy assessment. Although the method itself is not new, the contribution of this research would be able to provide a review of the meaning for the indices currently in use. The applicability of the method in practice is fairly low.
- This article is specifically targeted to DEM. However, the indices reviewed are for general use, not specifically for DEM. In practice, the uncertainty of the heights is considered to be related to the slope, and the land cover type. For a steep slope, the tolerance is larger. And for forest covered area is tolerance is also larger. This study seems overlooked this part.
- For the DEM in grid, the interpolation method and the density of the original point cloud have significant influence. These factors were also overlooked in this study.
- The current DEM production quality validation in the works this reviewer associated, the discrete point cloud is assessed first, then, the DEM in grid.
The authors are strongly suggested to re-organize the research for being related to DEM, if the DEM is the target.
Author Response
Dear reviewer, thank you very much for your comments and suggestions.
Q <<This article is specifically targeted to DEM. However, the indices reviewed are for general use, not specifically for DEM.>>
A: You are right, the standards presented in this paper are for general use, not specifically for DEM. A clarification has been included in the text in order to explain why DEMs are used.
Q <<In practice, the uncertainty of the heights is considered to be related to the slope, and the land cover type. For a steep slope, the tolerance is larger. And for forest covered area is tolerance is also larger. This study seems overlooked this part.>>
A: You are right. Also, there is a lot of literature indicating the relationships you point to.
In any case, most of the references on DEM evaluation do not proceed to a stratification. It is very usual to analyze all the altimetric errors as a whole. What is powerful and interesting about the applied mixing tool, as it is exposed in the discussion, is that the different normal distributions that make up the mixture can be related to what you indicate. This relationship that we intuit has not been demonstrated in this work, since it is not its objective and it would result in a very extensive document.
Q <<For the DEM in grid, the interpolation method and the density of the original point cloud have significant influence. These factors were also overlooked in this study. The current DEM production quality validation in the works this reviewer associated, the discrete point cloud is assessed first, then, the DEM in grid.>>
A: You are right.
However, we do not have the original point clouds.
We only have the two data products that are identified in the document. For these two products, the metadata offered by the producer regarding accuracy are presented.
The altimetric evaluation of a DEM product compared to other DEM products is very common in references. This is what we do, but our goal is not this. Our objective is to show the applicability of the finite mixtures method and to learn what happens if it is introduced as a statistical model in the current positional control standards.
Some clarifications have been included in the text
Q <<The authors are strongly suggested to re-organize the research for being related to DEM, if the DEM is the target.>>
A: As stated above, DEMs are not the target, the target are the mixtures and its application as base model for the PAAMs. DEMs offer an interesting and easier case to deal with.

Reviewer 3 Report
The authors undertook an important and interesting topic regarding the precision analysis of the DEM model.
However, the introduction lacks a description or mention of various methods of obtaining data for the needs of DEM and methods of generating this model. It is the source of the data and the method of generating the DEM that largely determines its accuracy, so I think that it should be supplemented.
A very well and solidly prepared description of the method - chapter 2. Finite mixture models. The only thing missing is the possibility of using it with the examples described in the literature (this is a suggestion, not a comment).
Fig 3. "Datos"? - translation in the drawing
Lines 394-397. "…. acceptable results regarding the distance between the obtained model (the fitted FMM) and the real data (the EDOD) ”. This definitely requires clarification and more precise values ​​as well as a broader analysis, it is a very important aspect of the work.
Line 449-451 "... models based on finite mixtures of normal distributions allow a better approximation to real cases of altimetric errors .." It requires the development and specification of a range of values ​​or at least presentation of the results of representative objects.
The manuscript is very well prepared and brings new information and touches upon important aspects of the accuracy of DEM. In my opinion, the manuscript after minor revision can be published in MDPI - Remote Sensing.
Author Response
Dear reviewer, thank you very much for your comments and suggestions. Thanks for your interest
Q <<However, the introduction lacks a description or mention of various methods of obtaining data for the needs of DEM and methods of generating this model. It is the source of the data and the method of generating the DEM that largely determines its accuracy, so I think that it should be supplemented.>>
A: Several comments have been added, also new references:
- David F. Maune, Amar Nayegandhi (2019). Digital Elevation Model Technologies and Applications: The DEM User’s Manual. ASPRS
- Höhle, J., Potuckova, M. (2011). Assessment of the Quality of Digital Terrain Models. Official Publication No 60 of European Spatial Data Research., Gopher, Amsterdam, The Netherlands, 92 p.
- Guth, Peter L., Adriaan Van Niekerk, Carlos H. Grohmann, Jan-Peter Muller, Laurence Hawker, Igor V. Florinsky, Dean Gesch, Hannes I. Reuter, Virginia Herrera-Cruz, Serge Riazanoff, Carlos López-Vázquez, Claudia C. Carabajal, Clément Albinet, and Peter Strobl. 2021. "Digital Elevation Models: Terminology and Definitions" Remote Sensing 13, no. 18: 3581. https://doi.org/10.3390/rs13183581
Q <<A very well and solidly prepared description of the method - chapter 2. Finite mixture models. The only thing missing is the possibility of using it with the examples described in the literature (this is a suggestion, not a comment).>>
A: Yes, you are right. For this reason, we have added some references about examples in section 5
Fig 3. "Datos"? - translation in the drawing
A: Corrected
Q<<Lines 394-397. "…. acceptable results regarding the distance between the obtained model (the fitted FMM) and the real data (the EDOD) ”. This definitely requires clarification and more precise values as well as a broader analysis, it is a very important aspect of the work.>>
A: An explanation has been included near Figure 5.
Q<<Line 449-451 "... models based on finite mixtures of normal distributions allow a better approximation to real cases of altimetric errors..." It requires the development and specification of a range of values or at least presentation of the results of representative objects.>>
A: The sentence has been rewritten.
We do not understand your comment at this point.
This sentence is a summary or conclusion of the work presented in the paper: The FMM works well, it adequately approximates the observed data, and allows working with the simplicity of a parametric model.

Reviewer 4 Report
The positional accuracy assessment of a DEM is very important and should always be aware before application. The popular assessment methods, like NMAS, EMAS, and NSSDA, based on either binomial distribution or normal distribution, usually require such statistics characteristic of the real error data. This could be difficult in many situations, due to the elevation measurements usually being non-stationary. The authors proposed the application of a finite mixture model to model altimetric errors, which is interesting. And it could be used in the elevation accuracy assessment of DEMs. Some comments are listed as follow, in order to further to improve the standard of the paper.
-First, the current abstract is too descriptive. It would be better to put some quantitative results or conclusions in this part, which can also highlight the results of this paper.
-Second, in the introduction, the authors stated a lot of why the research of positional accuracy assessment of DEM. However, little information has been paid to why the method of Gaussian finite mixture models. Actually, there are many methods available for the positional accuracy assessment of DEM, and it is necessary tell readers that why the authors used this method, instead of others. The inner relationship between the current problem of positional accuracy assessment and the Gaussian finite mixture models should be fully discussed in this paper. Otherwise, it is not enough to convince readers about the method.
-At the beginning of section 3, it is not necessary state a summary sentence of what do you do in this section. Instead, you can state the summary sentence of each section at the end of introduction section.
-I would like to suggest the author add a study area map in the paper. And the used DEM map can be also combined in this map. In addition, the basic information about the study area should be added. Thus, the readers can have a basic understanding of your study area.
-In addition, my concern also exists in the experimental data used in this paper. Although the real landform DEMs derived from the aerial LiDAR survey are necessary, the reference DEM with 2 m resolution still has uncertainty. Especially this paper concerning proposing a new error and positional accuracy standard, so the error-free surface, and controllable errors with different intensity are significant to this research. Hence, I suggest the authors can add relative experiments based on a mathematical surface. One can apply different densities and types of errors on this mathematical surface to explore and demonstrate the superiority of your proposed method. The mathematical surface can refer to the following papers.
Qin et al., (2007). An adaptive approach to selecting a flow-partition exponent for a multiple-flow-direction algorithm. International Journal of Geographical Information Science, 21(4), 443-458
Hu et al., (2021). Quantification of terrain plan concavity and convexity using aspect vectors from digital elevation models. Geomorphology, 375, 107553.
-Actually, there are different methods to evaluate the accuracy of DEM, like RMSE etc. some scholars also used terrain derivative (like slope etc.) or Shannon entropy (or configurational entropy) to evaluate the DEM. I am quite interesting the comparison between the current used method and the previous used methods. Some other methods of DEM evaluation can refer to the following papers.
Li et al., (2022). Integrating topographic knowledge into deep learning for the void-filling of digital elevation models. Remote Sensing of Environment, 269, 112818
Zhou et al., (2004). Analysis of errors of derived slope and aspect related to DEM data properties. Computers and Geosciences, 30(4), 369-378
-Finally, I suggest the author can publish the relative codes for the handle processes of the new method. It is helpful to the readers to reproduce the experiments and apply the proposed method to other fields.
Author Response
Dear reviewer, thank you very much for your comments and suggestions. Thanks for your interest
Q:<<First, the current abstract is too descriptive. It would be better to put some quantitative results or conclusions in this part, which can also highlight the results of this paper.>>
A: Some sentences of the abstract have been rewritten
Q: <<Second, in the introduction, the authors stated a lot of why the research of positional accuracy assessment of DEM. However, little information has been paid to why the method of Gaussian finite mixture models. Actually, there are many methods available for the positional accuracy assessment of DEM, and it is necessary tell readers that why the authors used this method, instead of others. The inner relationship between the current problem of positional accuracy assessment and the Gaussian finite mixture models should be fully discussed in this paper. Otherwise, it is not enough to convince readers about the method.>>
A: An explanation has been provided near the end of the introduction.
Q: <<At the beginning of section 3, it is not necessary state a summary sentence of what do you do in this section. Instead, you can state the summary sentence of each section at the end of introduction section.
A: This beginning has been rewritten
Q: <<I would like to suggest the author add a study area map in the paper. And the used DEM map can be also combined in this map. In addition, the basic information about the study area should be added. Thus, the readers can have a basic understanding of your study area.
A: A new figure has been added
Q: <<In addition, my concern also exists in the experimental data used in this paper. Although the real landform DEMs derived from the aerial LiDAR survey are necessary, the reference DEM with 2 m resolution still has uncertainty. Especially this paper concerning proposing a new error and positional accuracy standard, so the error-free surface, and controllable errors with different intensity are significant to this research. Hence, I suggest the authors can add relative experiments based on a mathematical surface. One can apply different densities and types of errors on this mathematical surface to explore and demonstrate the superiority of your proposed method. The mathematical surface can refer to the following papers.
- Qin et al., (2007). An adaptive approach to selecting a flow-partition exponent for a multiple-flow-direction algorithm. International Journal of Geographical Information Science, 21(4), 443-458
- Hu et al., (2021). Quantification of terrain plan concavity and convexity using aspect vectors from digital elevation models. Geomorphology, 375, 107553.>>
A: We consider that your comments are very interesting, however they are outside the scope of this work.
We do not propose a new method of evaluating positional accuracy.
The methods applied are the existing ones: the NMAS, EMAS and the NSSDA.
We only analyze the change in the statistical model underlying these methods.
That is, instead of continuing to use the model based on a single normal, use the calculation capabilities to fit a parametric model of finite mixture of normal distributions.
Q:<<Actually, there are different methods to evaluate the accuracy of DEM, like RMSE etc. some scholars also used terrain derivative (like slope etc.) or Shannon entropy (or configurational entropy) to evaluate the DEM. I am quite interesting the comparison between the current used method and the previous used methods. Some other methods of DEM evaluation can refer to the following papers.
- Li et al., (2022). Integrating topographic knowledge into deep learning for the void-filling of digital elevation models. Remote Sensing of Environment, 269, 112818
- Zhou et al., (2004). Analysis of errors of derived slope and aspect related to DEM data properties. Computers and Geosciences, 30(4), 369-378>>
A: We consider that your comments are very interesting, however they are outside the scope of this work.
For the use cases established by the NMAS, EMAS, and NSSDA methods, this paper proposes the use of a finite mixture model as a replacement for the model based on a single normal. The results obtained by the FMM, the 1NDM and the observed data are compared. This is what we focus on. We compare things that are comparable.
For example, in relation to the use of entropy. << It is a measure more related to certain properties of the frequency distribution but not directly related to the quality of the information in the layer (e.g. DEM).>> (Information entropy as a measure of DEM quality by Stephen Wise).
Q: <<-Finally, I suggest the author can publish the relative codes for the handle processes of the new method. It is helpful to the readers to reproduce the experiments and apply the proposed method to other fields.>>
A: thanks for the idea. Since this work has been financed through the FunQuality4DEm project, the data used in the research will be included on the website of this project (https://coello.ujaen.es/investigacion/web_giic/funquality4dem/ ).

Round 2
Reviewer 2 Report
1. As taught in Surveying, a compulsory course in Civil Engineering Department, the accuracy of digital terrain models is composed with several components. The most basic two would be the surveying accuracy and the slope of the terrain. That is, m(dem) =m1(method)+m2(tangent of slope). In the advanced course for Digital Elevation Modeling, there would be more components considered. A good textbook from my view would be the book published by CRC, https://www.routledge.com/Digital-Terrain-Modeling-Principles-and-Methodology/Li-Zhu-Gold/p/book/9780415324625 In the Chapter 8, the accuracy of DEM is comprehensively addressed. 2. For the paper under review, the theme is applying a well established statistical model for analysis. Although well established, it would be not well known for most readers. There would be a value, may not be significant, for publishing this paper to provide readers thinking about this issue. The statements in my original review report were with the intention that the authors could address the conventional DEM accuracy assessment indices, not just general mapping accuracy indices, in their writing. Or, even better, they could fuse these two schemes. The authors responded in their current revision by adding two references without any insight. I am neutral for accepting or rejecting. For reference to the Editor.